Registered report  

acoustics/psychology

hearing loss, reverberation, speech intelligibility

**Author for correspondence:**
Raphael Cueille
e-mail: raphael.cueille@entpe.fr

# Effects of reverberation on speech intelligibility in noise for hearing-impaired listeners

Raphael Cueille[1,2], Mathieu Lavandier[1] and Nicolas Grimault[2]

[1]Univ. Lyon, ENTPE, Ecole Centrale de Lyon, CNRS, LTDS, UMR5513, Vaulx-en-Velin 69518, France
[2]CRNL, UMR CNRS 5292, Univ. Lyon 1, 50 av T Garnier, Lyon Cedex 07 69366, France

RC, 0000-0003-1308-0591

Reverberation can have a strong detrimental effect on speech intelligibility in noise. Two main monaural effects were studied here: the temporal smearing of the target speech, which makes the speech less understandable, and the temporal smearing of the noise, which reduces the opportunity for listening in the masker dips. These phenomena have been shown to affect normal-hearing (NH) listeners. The aim of this study was to understand whether hearing-impaired (HI) listeners are more affected by reverberation, and if so to identify which of these two effects is responsible. They were investigated separately and in combination, by applying reverberation either on the target speech, on the noise masker, or on both sources. Binaural effects were not investigated here. Intelligibility scores in the presence of stationary and modulated noise were systematically compared for both NH and HI listeners in these situations. At the optimal signal-to-noise ratios (SNRs) (that is to say, the SNRs with the least amount of floor and ceiling effects), the temporal smearing of both the speech and the noise had a similar effect for the HI and NH listeners, so that reverberation was not more detrimental for the HI listeners. There was only a very limited dip listening benefit at this SNR for either group. Some differences across group appeared at the SNR maximizing dip listening, but they could not be directly related to an effect of reverberation, and were rather due to floor effects or to the reduced ability of the HI listeners to benefit from dip listening, even in the absence of reverberation.

# 1. Introduction

In our daily lives, we often experience situations where we have to understand speech hindered by an interfering noise. In these conditions, reverberation can greatly decrease our speech understanding abilities. This study aims to better understand

how reverberation could affect intelligibility for hearing-impaired (HI) listeners. This should allow to understand more precisely which acoustic conditions are detrimental to them and, in the long term, it could help to improve the acoustic regulations related to room design. For example, if the speech intelligibility of HI listeners is reduced at lower levels of reverberation compared with normal-hearing (NH) listeners, more restrictive reverberation limits could be recommended in public buildings.

Speech intelligibility in a noisy environment depends on several factors [1,2] that can be influenced by reverberation. It has been shown that in a room the envelope of the sound can be temporally smeared, depending of the level of reverberation [3], due to the individual soundpaths with various time delays taken by the sound reflections. The temporal smearing of the target speech has been shown to reduce its intelligibility [4,5]. When speech is heard in the presence of an interfering noise, speech intelligibility increases if the noise is modulated in amplitude [6]. NH listeners seem to be able to benefit from the 'gaps' in the modulated noise to better understand the target speech. This benefit is often called 'dip listening advantage', 'benefit of listening in the gaps' or 'fluctuating masker benefit'. In a room, the dip listening benefit seems to be less important because reverberation fills in the masker gaps. For example, a recent study measured speech reception thresholds (SRTs, sound level of the target compared with that of the interferer to recognize 50% of the words) for NH listeners with interfering noise and different degrees of reverberation applied to the noise [7]. The noise was either stationary or modulated using various speech envelopes, while the level of reverberation was changed by varying the distance between the noise source and the listener. When increasing reverberation was applied to a modulated noise, the dip-listening benefit was significantly reduced. In addition to these two monaural mechanisms, it has been shown that speech intelligibility is improved when the noise source is at a different position than the target speaker, thanks to our binaural system [8]. This advantage from binaural hearing is also reduced by reverberation [9,10]. As a starting point, the present study focuses on the two monaural effects of reverberation, so that binaural effects were not investigated here. It has also been shown that auditory stream segregation and attentional effects are less affected by reverberation [11]. These effects were not studied here. Since stream segregation effects can be very important with speech interferers, only noise interferers were used in the present study.

A few previous studies have compared the monaural effects of reverberation for NH and HI listeners. Duquesnoy & Plomp [12] tested one group of NH listeners and four groups of HI listeners with different degrees of hearing loss. When increasing the level of reverberation as measured by the reverberation time (RT), from 0 to 2.3 s, the SRTs of the NH listeners increased by 10 dB. The SRTs of the most severely impaired HI listeners increased by 18 dB when going from RT = 0 s to RT = 1 s. However, since the study had a different aim, the effect of reverberation on SRTs was not statistically analysed, so the significance of the effect remains unknown.

Helfer & Wilber [13] tested four groups of listeners, old and young NH and HI listeners, at four different RTs, with and without noise. Overall, this study reported a decrease in intelligibility scores (the percentage of words correctly identified) for all groups when reverberation increased and a significant interaction between the group of listeners and the reverberation time. In more detail, applying reverberation to both the target speech and the (cafeteria) noise decreased the intelligibility scores by 20% for the NH groups and by 25% for the HI groups. Both temporal smearing of the target and a reduction in dip listening could account for these results. An additional condition without making noise also evidenced a detrimental effect of reverberation with a 10% drop in intelligibility for the NH listeners and a 20% drop for the HI listeners. However, when expressed in percentage, the results can be biased by floor and ceiling effects. To correct this, it is necessary to convert the scores into rationalized arcsine units (RAUs) [14,15]. When the scores from Helfer & Wilber [13] are converted into RAUs, the differences between groups are reduced. NH listeners have intelligibility decreases of 22 and 20 RAUs for the 'speech in noise' and the 'speech in quiet' conditions, respectively, while the HI listeners have intelligibility decreases of 28 and 25 RAUs. This study did not investigate at which reverberation level speech intelligibility decreased the most for the different groups.

Harris & Swenson [16] compared a group of NH listeners with two groups of HI listeners: one group with mild hearing loss and another group with severe hearing loss. In this study, intelligibility scores were measured at three levels of reverberation, in the presence of a stationary noise. The results showed that reverberation caused a larger decrease in scores for the HI groups (30% decrease for the HI groups against 20% decrease for the NH group). They also found that HI listeners were more affected than NH listeners for lower RTs. The intelligibility decrease for RTs increasing from 0 to 0.5 s was equal to 10% for the NH listeners against 20% for the HI listeners. The intelligibility decrease for RTs increasing from 0.5 to 1.5 s was equal to 12% for the NH listeners and 8% for the HI listeners.

Since the interfering noise was stationary, no listening in the gaps was involved in any condition, and it can be assumed that the effect of reverberation was restricted to the effect of target smearing. This could imply that HI listeners are more affected by temporal smearing than NH listeners. This could also indicate that HI listeners are affected at lower reverberation levels, compared with NH listeners. However, these results could also be partly explained by a ceiling effect: when the intelligibility scores are converted in RAUs, the intelligibility decrease when the RT increases from 0 s to 1.5 s is of 28 RAUs for the NH listeners and 32 RAUs for the HI listeners. Between 0 and 0.5 s, the intelligibility decrease is of 15 RAUs for the NH listeners and 22 RAUs for the HI listeners. These results are also not consistent with another study, which tested NH and HI listeners at two low reverberation levels (RT = 0.25 and 0.5 s) [17]. Both groups had significantly lower intelligibility scores at the higher reverberation level. However, while the NH group had better performances overall, the intelligibility decrease was more important for the NH group than for the HI group (29 RAUs and 16 RAUs, respectively). This inconsistency between the two studies could perhaps be linked to the difference in the hearing losses tested: the former study had two HI groups with different levels of hearing loss (HL), corresponding to a mean pure tone average (PTA) loss of 30 and 60 dB HL, respectively; while the later study had HI listeners with heterogeneous losses, with PTA ranging from 13 to 70 dB HL. It could also come from a difference in room geometry and/or source distances. Both studies used rooms of similar sizes (11.92 m$^3$ [16] and 13.6 m$^3$ [17]) but the first room had non-parallel walls while the second had parallel walls. Unfortunately, the direct to reverberant energy ratios (DRRs) were not provided in these studies, so nothing can be concluded here. It is also worth noting that the first study had a total of 30 listeners (10 NH listeners and 20 HI listeners), while the second only had 12 (5 NH listeners and 7 HI listeners). It remains unclear whether HI listeners are more affected by reverberation than NH listeners and at which reverberation levels they are affected the most. Many of the differences between NH and HI listeners reported in the literature have not been verified by statistical tests and could have been biased by ceiling effects.

To the best of our knowledge, the decrease of dip listening benefit caused by reverberation in the case of HI listeners has only been investigated by one study [18]. However, the HI listeners did not have much dip listening benefit, even in the anechoic condition, and no statistical test was done regarding this particular interaction. It is still possible to make some assumptions, based on numerous studies on dip listening benefit for HI listeners without reverberation. When tested at different signal-to-noise ratios (SNRs, sound level of the target compared with that of the interferer) providing comparable SRTs for NH and HI listeners, HI listeners generally have very poor dip listening benefit. For example, Bernstein & Grant [19] have estimated SRTs for 20 NH and 20 HI listeners with various types of masker (stationary noise, modulated noise, competing voice). They found that the NH SRTs were 4–8 dB lower with a modulated noise compared with a stationary noise, while HI listeners did not benefit at all from the masker modulations. This seems to imply that HI listeners cannot rely on dip listening as much as NH listeners do. Hall et al. [20] measured the SRTs of NH and HI listeners using stationary and temporally modulated noises. They found a significantly higher dip listening benefit for the NH listeners. Lorenzi et al. [21] measured consonant identification scores of NH and HI listeners using noise maskers with amplitude modulation rates varying from 2 to 128 Hz. They found no dip listening benefit for half of the HI listeners, while the other half had a smaller benefit when compared with the NH listeners.

The reduced dip listening benefit of HI listeners could be partly due to the SNRs at which they were tested. It has been shown that the dip listening benefit depends on the SNR at which it is measured: when the SNR increases, the dip listening benefit decreases [22]. When tested at the same SNR (thus generally not at the same intelligibility performance), NH and HI listeners have much more comparable dip listening benefit [19]. Further studies controlled for the effect of SNR on dip listening. One tested HI and NH listeners at frequencies at which they had similar hearing thresholds and found no significant difference in dip listening benefit between groups [23]. In another study, the HI listeners were tested with a closed set procedure while the NH listeners were tested with an open set procedure [24]. The scores from the NH listeners were reduced and both groups had similar scores at the same SNRs. Since the main goal of this study was to study the effects of supra-threshold deficits on the dip listening benefit, the audibility of both groups was equalized by filtering the stimuli, to eliminate the elements below the hearing thresholds of the HI listeners. This study found no significant difference in dip listening benefit between groups, except at a high modulation rate (32 Hz). The authors concluded that when HI listeners were not impaired by their reduced audibility, they were able to benefit from dip listening, except at high modulation rates where they might be hindered by a reduced temporal resolution.

If HI listeners can at least partly benefit from dip listening, it is relevant to assume that HI listeners might be affected by the decrease of dip listening caused by reverberation. Studies which measured the intelligibility scores of HI and NH listeners with square-wave modulated noise might provide some insights to test for this hypothesis. For example, George *et al.* [25] have tested the effect of various duty cycles (i.e. the fraction of time period during which the noise is present) upon intelligibility. The reduction of this duty cycle can be seen as a basic representation of the width reduction of the noise gaps caused by reverberation. It appeared that HI listeners had lower scores overall and the score decrease caused by the duty cycle reduction was greater for the NH listeners. This could indicate that HI listeners do not benefit as much from dip listening as NH listeners do, and thus the effect of reverberation on the dip listening benefit might be more important for NH listeners than for HI listeners.

The aim of the present study was to better understand the effect of reverberation on speech intelligibility in noise for HI listeners. The effects of target temporal smearing and the reduction in dip listening were addressed separately as well as in combination, for both HI and NH listeners. Based on the literature, this study tested the following hypotheses:

  (i) Hypothesis 1: the temporal smearing of the speech caused by reverberation impairs more HI listeners than NH listeners.
 (ii) Hypothesis 2: reverberation affects the dip listening benefit more for NH listeners than for HI listeners.
(iii) Hypothesis 3-1: the overall (monaural) effect of reverberation impairs HI listeners as much as NH listeners.

In addition to these three main hypotheses, this study also aimed to test several secondary ones:

  (i) Hypothesis 1-1: the temporal smearing of the target speech impairs intelligibility at lower levels of reverberation for HI listeners compared with NH listeners (that is to say HI listeners are more affected than NH listeners at low RTs).
 (ii) Hypothesis 1-2: at high levels of reverberation, HI listeners are more impaired than NH listeners by the temporal smearing of the speech (that is to say, HI listeners are more affected than NH listeners at high RTs).
(iii) Hypothesis 3-2-a: in the case of NH listeners the dip listening decrease has a significant effect on intelligibility when reverberation is applied to speech and noise.
 (iv) Hypothesis 3-2-b: In the case of HI listeners the dip listening decrease has a significant effect on intelligibility when reverberation is applied to speech and noise.

To test these hypotheses, it is necessary to measure speech intelligibility for HI and NH listeners, applying reverberation either only on the target speech, only on the interfering noise, or on both sources, as previously done for NH listeners [26]. This allowed to address separately two effects of reverberation: the temporal smearing of the target and the reduction in dip listening. Care was also taken to assure that the dip listening benefit was evaluated at comparable SNRs for HI and NH listeners.

# 2. Methods

## 2.1. Listeners

The 32 NH listeners were aged between 55 and 72 years old (mean age of 62 years old, standard deviation of 5 years). The 32 HI listeners were aged between 55 and 79 years old (mean age of 67 years old, standard deviation of 7 years). The NH listeners had hearing thresholds below 20 dB HL from 125 to 4000 Hz and below 25 dB HL from 6000 to 8000 Hz. Their best ear was selected for testing. The HI listeners had mild-to-moderate hearing loss. The four-frequency (0.5, 1, 2, 4 kHz) average hearing loss (4FAHL) and ±1 s.d. was 12.6 ± 3.3 dB HL for the NH listeners and 45.2 ± 11.3 dB HL for the HI listeners. The individual audiograms of the NH and HI listeners at the tested ear as well as the mean audiograms across listeners can be found in figure 1.

The study received formal agreement from a French national ethic committee (CPP Ile de France X, protocol no. 05-2019). The listeners signed an informed-consent form and were paid an hourly wage for their participation.

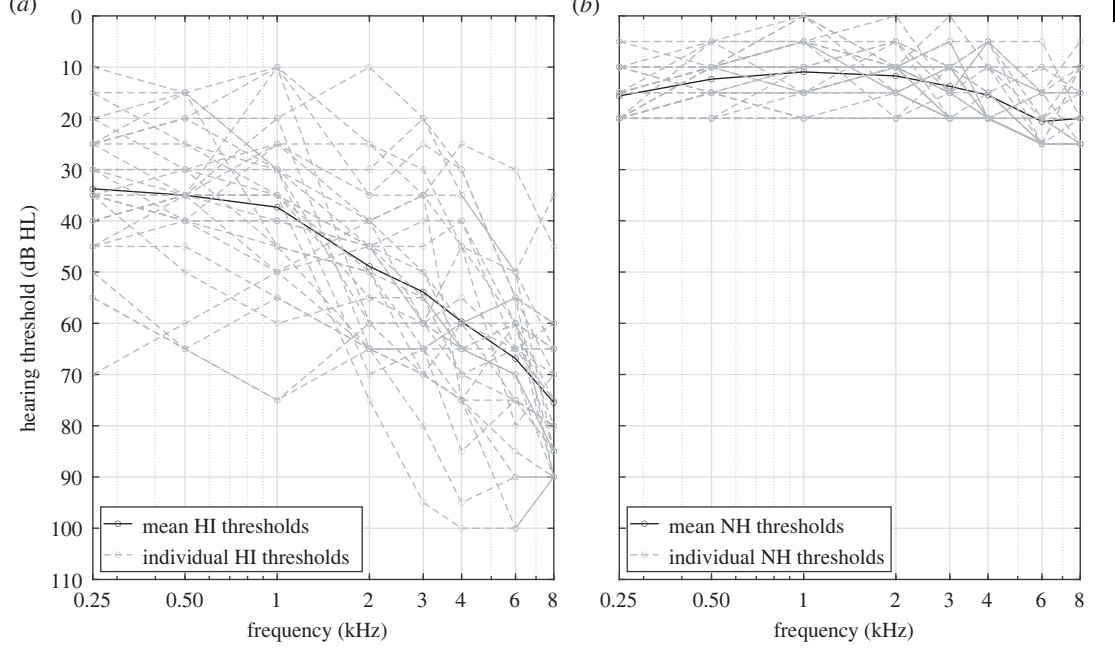

**Figure 1.** Audiograms of the HI (*a*) and NH (*b*) listeners at the tested ear. Each listener is represented by a grey dotted line. The black solid lines represent the average across listeners.

## 2.2. Stimuli

A matrix-style closed-set speech material was used. It allows to prepare as many sentences as required for the large number of conditions to be tested, from a limited number of words. At the same time, the difficulty across sentences is rather uniform for such material, unlike in open-set tests where difficulty can vary substantially across sentences. This minimizes variability due to the materials, which might otherwise contribute to the variability across individuals and tested conditions that are of primary interest here. The speech material of the French Matrix test (FrMatrix) was used [27]. This closed test consists of five groups of 10 words: 10 names, 10 verbs, 10 numerals, 10 objects and 10 colours. These words can be arranged to form various sentences, for example 'Agnès attrape cinq crayons jaunes' (Agnes catches five yellow pencils) or 'Etienne déplace six jetons bleus' (Etienne moves six blue chips). The number of words recognized by the listener is then measured. The words were chosen so that they best represent French mean phonetic distribution. Jansen *et al.* [27] created 280 sentences with these words. The recording of these sentences was done in a way that allowed to preserve the coarticulation between words. The sound levels of the words were calibrated so that there was equal intelligibility across words, sentences and test lists. To preserve this equalization in the experiment, only the mean level across all sentences from all the lists was varied. The 50 words composing the test can be found in table 1.

To study the effect of reverberation on the dip listening benefit, two types of noise were tested: a stationary noise and a modulated noise. The dip listening benefit was estimated as the score difference between the condition with modulated noise and the condition with stationary noise at a given SNR. For the modulated noise, 288 semantically unpredictable sentences from another speech material [28] were used to create two one-voice vocoded speeches (of 144 sentences each). These two vocoded speeches were then added to create a two-voice vocoded speech. The use of this speech material allows to have less predictable gaps than with the sentences of the FrMatrix, which all have the same structure and length. A three-channel vocoder was used, with an envelope low-pass filter-cut off frequency of 60 Hz [29]. A finite impulse response (FIR) filter was used to match the excitation pattern of the vocoded speech with one of the target sentences so that they had the same long-term spectrum. Speech modulations were used instead of artificial modulations (such as sinusoidally modulated noise) to avoid any effect of the predictability of the gaps in the masker [7]. The stationary noise was created by concatenating the same semantically unpredictable sentences, so that it had the same long-term spectrum as the modulated noise. The Fourier transform of the concatenated signal was calculated, its phase was randomized and the inverse Fourier transform was taken [7]. Similarly to the

**Table 1.** Words of the FrMatrix test [27].

| name | verb | numeral | object | colour |
|------|------|---------|--------|--------|
| Agnès | achète | deux | anneaux | blancs |
| Charlotte | attrape | trois | ballons | bleus |
| Emile | demande | cinq | classeurs | bruns |
| Etienne | déplace | six | crayons | gris |
| Eugène | dessine | sept | jetons | jaunes |
| Félix | propose | huit | livres | mauves |
| Jean-Luc | ramasse | neuf | pions | noirs |
| Julien | ramène | onze | piquets | roses |
| Michel | reprend | douze | rubans | rouges |
| Sophie | voudrait | quinze | vélos | verts |

**Table 2.** Absorption coefficients ($\alpha$, in %) of the simulated rooms, along with the RT (in seconds) and DRR (in decibel) evaluated between the source and ear positions.

| RT | DRR | $\alpha$ (125 Hz) | $\alpha$ (250 Hz) | $\alpha$ (500 Hz) | $\alpha$ (1 kHz) | $\alpha$ (2 kHz) | $\alpha$ (4 kHz) |
|----|-----|----------|----------|----------|---------|---------|---------|
| 0.15 | 10.7 | 58 | 69 | 96 | 97 | 99 | 91 |
| 0.5 | −6.05 | 17.4 | 20.7 | 28.8 | 29.1 | 29.7 | 27.3 |
| 0.8 | −8.97 | 8.7 | 10.35 | 14.4 | 14.55 | 14.85 | 13.65 |
| 1.1 | −11.0 | 5.8 | 6.9 | 9.6 | 9.7 | 9.9 | 9.1 |
| 1.5 | −12.21 | 3.77 | 4.485 | 6.24 | 6.305 | 6.435 | 5.915 |

vocoded speech, the stationary noise was filtered so that it had the same long-term spectrum as the target sentences. Random excerpts of these two long noises were selected and gated on and off with 10 ms raised-cosine ramps. Each excerpt was one second longer than its affiliated target and began between 300 and 700 ms before the target sentence, the exact delay being randomly selected on each trial.

To add reverberation to the noise and/or to the target speech, several monaural room impulse responses (RIRs) were artificially generated using the software CATT-acoustic (v. 9.1). This software enables simulation of the sound reaching a microphone from a source, with the source and microphone simulated at any positions within the room, with adjustable geometric and acoustic properties of the walls, ceiling and floor. The room was created with the following dimensions: 5 × 9 × 2.7 m. The source was placed at 1.75 m from the 5 m wall and 3.75 m from the 9 m wall. The listener was placed at 3.25 m from the 5 m wall and 5 m of the 9 m wall. These positions were chosen so that the source and listener were not too far from each other and far away from the corners of the room. Both sources were placed at a height of 1.5 m to simulate standing persons. To control the reverberation level using a single parameter, the floor, ceiling and all the walls were given the same frequency-dependent absorption coefficients of mineral wool. To increase the reverberation level to reach a given broadband RT, these coefficients were decreased simultaneously in all frequency bands by division (table 2). With this configuration, the only varying parameter was the gain applied to the absorption coefficient of the rooms. This allows a precise control of the reverberation: between each RIR only the mean level of the reflections changes, preserving the relative level and delay differences across reflections. All the RIRs were normalized in energy, so that convolution did not introduce broadband level differences across reverberation conditions. The mean presentation level of the speech sentences across all tested lists was calibrated at 50 dB sound pressure level (SPL) and the noise level was adapted to reach the desired SNR in each tested condition.

Modulation spectrum analyses were done to determine which RTs were most interesting to test. The modulation spectra of the noise and speech signals were generated by computing their envelopes,

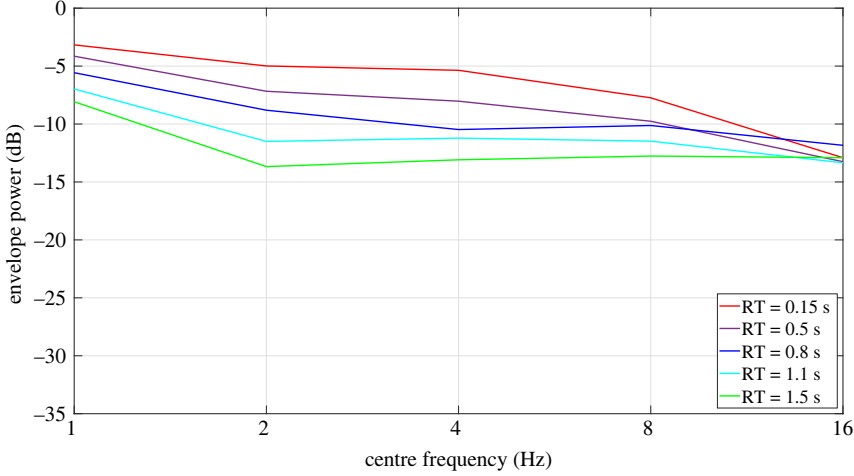

**Figure 2.** Modulation spectra of the speech signals convolved with the RIRs of the virtual room for different RTs between 0.15 and 1.5 s.

dividing it by its mean and calculating the power spectrum in octave bands between 1 and 16 Hz, as done by Collin & Lavandier [7]. The modulation spectra of the selected conditions are presented in figures 2 (speech) and 3 (noises). The peak that can sometimes be observed at low frequencies is due to an artefact inherent to the room simulation program, which creates a DC component increasing with the level of reverberation, but this DC component is not audible, and it was discarded in the level calculation used for the equalization. The condition RT = 0.15 s used very high absorption coefficients, a proxy for testing almost anechoic conditions.[1]

For the speech signals, the effect of reverberation at low reverberation times was investigated (hypothesis 1-1) as well as the potential interaction of the effect of reverberation with the listener group (hypotheses 1, 1-2), so it was necessary to test low and high reverberation times. The modulation spectra of the five reverberation times that were selected appear in figure 2. When reverberation increases the speech signals show a decrease in modulation level.

For the noise signals, only the presence of an interaction of the effect of reverberation with the listener group (hypotheses 2, 3-1, 3-2-a and 3-2-b) was investigated, so it was more relevant to test only two extreme levels of reverberation. As a result, only the RT values of 0.15 s and 1.5 s values were used for the noise signals. Figure 3 shows a decrease of noise modulation level (for the modulated noise) with increasing reverberation. The acoustic properties of the artificial rooms tested are presented in table 2.

It was necessary to ensure that there was at least one common SNR for all listeners that allows for a dip listening effect and at the same time that there was no floor or ceiling effect for the two groups at this SNR. To ensure the existence of such an SNR, HI listeners had linear frequency-dependent gain applied to the stimuli before presentation. This gain was set on an individual basis according to the NAL-RP prescription rule [30]. This linear amplification allowed to partly compensate for the audibility loss and ensured that the score difference at the same SNR was not too large between the HI and NH groups. It was decided to not completely equalize the audibility of the HI and NH listeners, in order not to focus only on the supra-threshold elements of the hearing loss. This method avoids the confounding effect of tested SNR without being affected by floor and ceiling effects. At the same time it still facilitates testing for a difference of audibility between HI and NH listeners.

## 2.3. Procedure

All the listening tests were done monaurally in a sound-proof booth (sound attenuation over 70 dB from 100 Hz to 20 kHz) with headphones (Sennheiser HD 250 Linear II). All sound levels were calibrated with an artificial ear associated with a flat coupler (Larson Davis AEC101 and 824).

The stimuli on each trial were the five words of the target sentence and an interfering noise at a given SNR. On each trial, listeners chose an answer for each of the five words. These choices were made by

[1]Our modulation analyses showed little difference between the anechoic condition and the RT = 0.15 s condition. It was decided to keep the RT = 0.15 s so that all the signals were convoluted with RIRs varying only due to a single parameter (the gain applied to the absorption coefficients of the room).

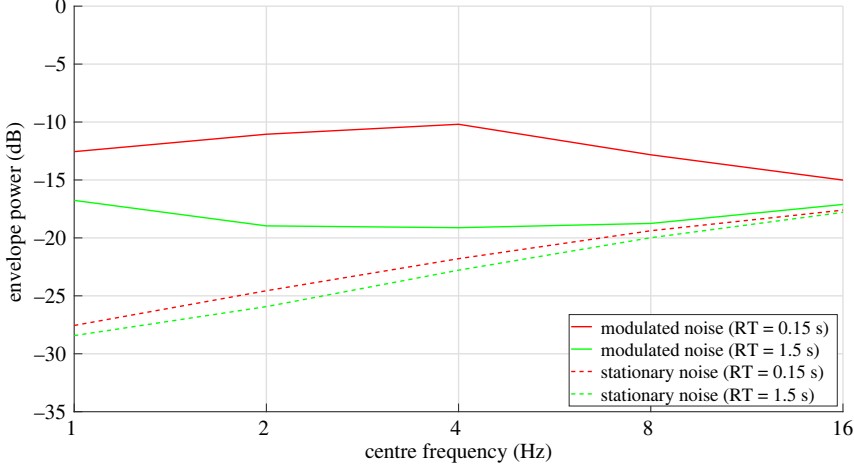

**Figure 3.** Modulation spectra of the modulated and stationary noises convolved with the RIRs corresponding to the two extreme RTs (0.15 and 1.5 s).

selecting among all the possible words presented on a computer screen in front on them. Matrix tests are often used with 20 sentences that use twice the 50 words of the speech set to do an adaptive measure of the SRT [27]. In this experiment, to avoid the confounding effect of SNR on the dip listening benefit [22], it was decided to measure the intelligibility scores at five SNRs for each listener. For each tested SNR, participants listened to 10 sentences that contained the 50 words of the FrMatrix (in other words, for each condition 50 sentences were tested to obtain the psychometric function). The words layout and the tested conditions were selected in random order on a sentence-by-sentence basis.

For each listener and condition, psychometric functions were then fitted to the percentage scores,[2] with the psignifit toolbox [31]. Each psychometric function was modelled with a cumulative Gaussian sigmoid curve $\psi$(SNR) parametrized by its threshold $m$ (inflexion point of the function) and width $w$ (range of SNR containing 95% of the function) and scaled by upper and lower asymptotes $\lambda$ and $\gamma$. The variables $m$ and $w$ were estimated with a maximum-likelihood method, as described by Wichmann & Hill [32]. $\lambda$ was allowed to vary between 0 and 0.05, to take into account the listeners' lapses in performance. The sigmoid function used is detailed in equation (2.1).

$$\psi(\text{SNR}) = \gamma + \frac{(1 - \gamma - \lambda)}{(1 + \text{e}^{-2*\log(1/0.05-1)/w*(\text{SNR}-m)+\log(1/0.5-1)})}.\tag{2.1}$$

The individual fitted psychometric curves were then converted from percentage to rationalized arcsine units (RAU)[3] [14] in order to do the statistical analyses. The mean psychometric functions of the listeners were also created to visualize the results per group. These mean curves were created by averaging across listeners the parameters ($\lambda$, $\gamma$, $m$ and $w$) of the individual psychometric functions within each group. The curves that are presented in the article (figures 4–6) are the RAU transformation of the mean psychometric functions of the listeners. The mean SRTs of each group were also calculated and can be found in the electronic supplementary material.[4]

The NAL-RP amplification applied to the stimuli for the HI listeners increased the sound level by up to 20 dB for the most impaired listeners. With the loudest stimuli having a noise sound level of 70 dB, the NAL-RP amplification could lead to stimuli presented at 90 dB SPL. To make sure that the stimuli were never uncomfortable for the HI listeners, a loudness test was done at the beginning of the experiment, with a method similar to the Contour test [33]. The listeners were presented with the stimuli (a sentence and a modulated noise at the highest SNR) and had to evaluate their loudness on a 7-point scale: 1, very soft; 2, soft; 3, comfortable but slightly soft; 4, comfortable; 5, comfortable but slightly loud; 6, loud, but okay; 7, uncomfortably loud. The sound level of the stimuli started at 55 dB and was raised by steps of 5 dB until the listener found the stimuli uncomfortably loud. If it was

---

[2]This methodological choice was made before data observation.

[3]The RAU transformation was not done on the raw scores measured during the experiment, but on the fitted psychometric curves.

[4]See electronic supplementary material at https://zenodo.org/record/6542359#.YnzkJ-hByUk.

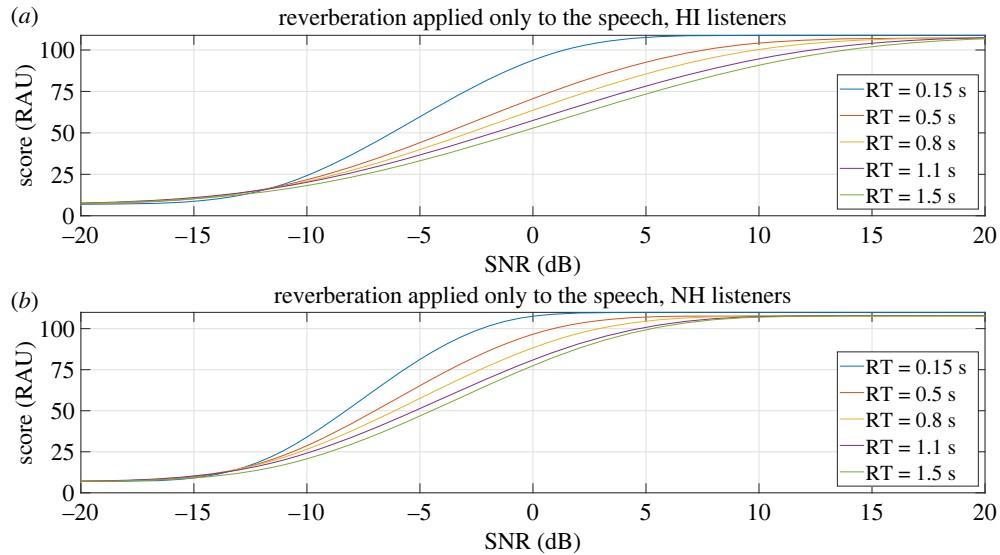

**Figure 4.** RAU conversion of the mean psychometric curves of the HI (*a*) and NH (*b*) listeners when reverberation was applied only to the speech.

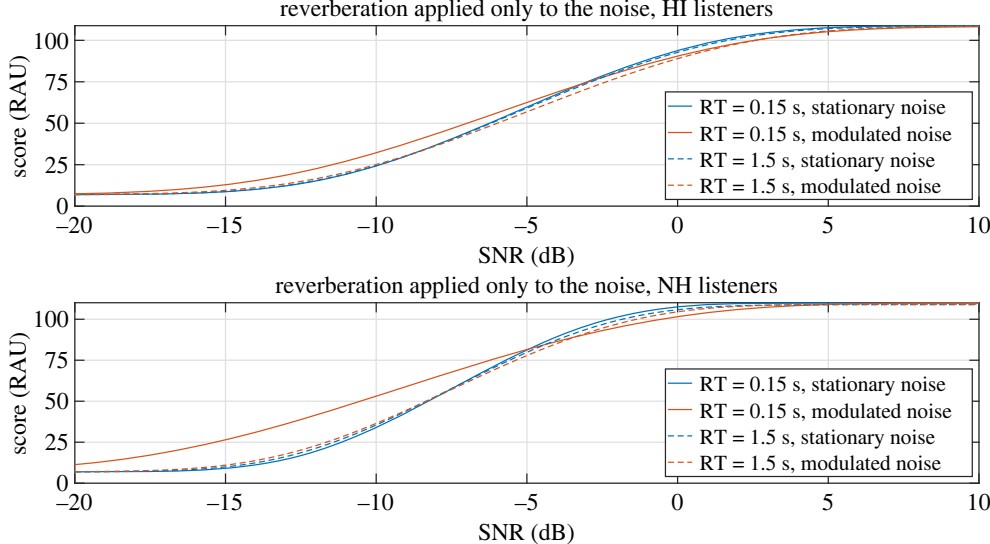

**Figure 5.** RAU conversion of the mean psychometric curves of the HI and NH listeners when reverberation was applied only to the noise.

necessary, in the intelligibility test, the SNR range was adjusted so that the listener was always tested below the sound level judged uncomfortably loud.

The low 50 dB SPL presentation level used for the target speech in the intelligibility experiment limited the risks of discomfort. However, it raised the question of the audibility of the sentences. To make sure that they were audible in quiet, an audibility test was done before the loudness test. It consisted in measuring the intelligibility for a list of 20 sentences presented in quiet. For six HI listeners and four NH listeners, the intelligibility score in this quiet condition was below 90%. For these listeners, the presentation level was raised by steps of 5 dB and the audibility test was redone until the intelligibility score was above 90%. The presentation level was raised by 10 dB for one HI listener and by 5 dB for the four other HI listeners and for the four NH listeners.

Each listener was tested during three sessions (for more details, see the next section). Each listening session was preceded by a practice test consisting of two 20-sentence lists, so that listeners got familiar with the task and to limit the short-term training effect that comes with a closed-set speech test [27]. There also seems to be a training effect in the long term, across multiple sessions [34]. It does not

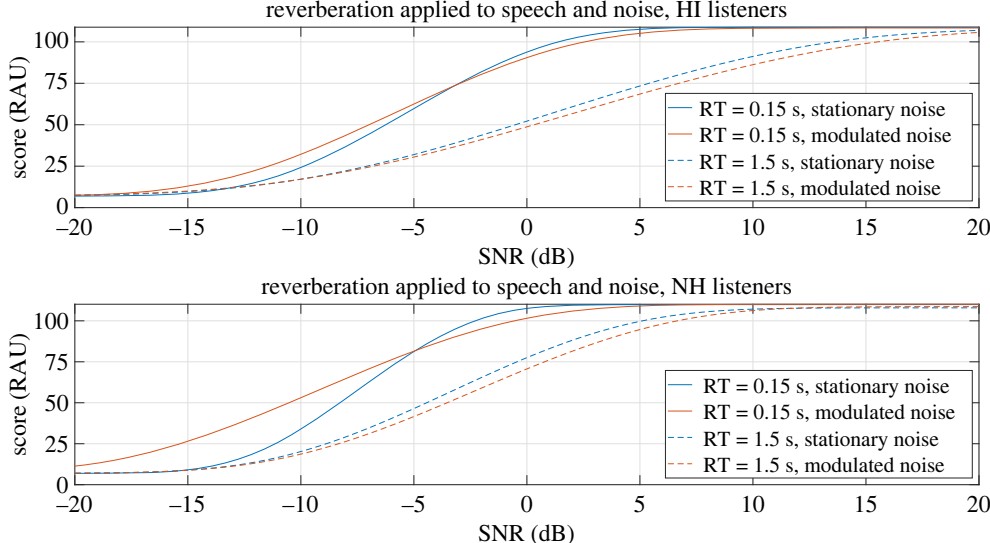

**Figure 6.** RAU conversion of the mean psychometric curves of the HI and NH listeners when reverberation was applied to both speech and noise.

**Table 3.** Conditions tested in the present study.

| application of reverberation | speech only | noise only | speech and noise |
| --- | --- | --- | --- |
| underlying mechanism tested | temporal smearing | dip listening | both |
| RTs | 0.15, 0.5, 0.8, 1.1, 1.5 s | 0.15, 1.5 s | 0.15, 1.5 s |

seem possible to compensate for this effect, since no study determined the training effect for this speech set in terms of intelligibility percentages for the SNRs tested in the present study. This might have added some variability to the measures but should not affect the differences measured between conditions, since they were randomly tested at different moments of the experiment. Another drawback of matrix speech sets is that they seem less affected by reverberation than daily life speech [35]. Such an effect could not be compensated for and is addressed in more details in the Discussion.

## 2.4. Tested conditions

To test the hypotheses of this study, three ways of applying reverberation to the stimuli were tested for each listener: one with reverberation applied to both speech and noise (both temporal smearing and a decrease in dip listening could impair intelligibility), one with reverberation applied only to the speech (only temporal smearing could contribute to the effect of reverberation) and one with reverberation applied only to the noise (only a decrease in dip listening could contribute to the effect of reverberation). For each reverberation condition, different RTs were tested, for both types of noise modulations (stationary, speech-modulated). These conditions are summarized in table 3.

The condition with reverberation applied only to the target speech tested the hypothesis that temporal smearing reduces speech intelligibility at low RTs for HI listeners and at higher RTs for NH listeners. To do so, it was necessary to test more RTs for this condition. Based on previous work [16] and on the modulation analyses (figure 1), the following RTs were selected: 0.15, 0.5, 0.8, 1.1, 1.5 s. Reverberation was applied only to the target, so that dip listening was not affected by reverberation [7].

The condition with reverberation applied only to the noise tested for an interaction between the effects of listener group (HI versus NH), noise modulations and level of reverberation. The idea here was to test whether the dip listening benefit is affected differently by reverberation for the two groups of listeners. Since this hypothesis studied the presence of an interaction, the two types of noise were only tested at the two extreme RTs. For this condition, reverberation was only applied to the interfering noise, so that there was no temporal smearing of the target speech.

The condition with reverberation applied to both sources tested the hypothesis that reverberation (overall) does not affect differently NH and HI listeners. Similarly to the condition with reverberation applied only to the noise, this hypothesis studied the presence of an interaction, and only the two extreme levels of reverberation (RTs of 0.15 and 1.5 s) were applied simultaneously to noise and speech.

Each combination of reverberation condition × RT × noise modulation was tested at five SNRs in order to measure the full psychometric function of the participants. This makes a total of 70 conditions. Each condition was measured in approximately 2 min (12 s per sentence), which makes a total of 2 h and 20 min for the whole experiment. The experiment was done in three sessions of 1 h and 15 min each, in order to have the time to measure the audiograms and realize the loudness, intelligibility in quiet and practice tests. To avoid the risk of memorization, the tested conditions—SNR, RT, masker modulation, reverberation condition—were selected in random order on a sentence-by-sentence basis.

## 2.5. Data analysis

### 2.5.1. Statistical analyses

All the analyses were done at a single common SNR of the RAU conversion of the individual psychometric curves of the listeners. This SNR was chosen so that ceiling and floor effects were kept to a minimum. Our analyses primarily rely on Bayesian ANOVAs. Bayesian tests not only allow to show evidence for the presence of an effect, but also for its absence, something that frequentist tests do not permit [36]. In our case, this was particularly useful to study the interaction of dip listening and reverberation, for which it was more difficult to anticipate whether there would be differences between the HI and NH groups. This also allowed to study the hypothesis 3-1 for which we intended to show the absence of an effect.

Five Bayesian-mixed-design-repeated-measures ANOVAs were planned, one for each reverberation condition (applied to speech, to noise or to both) and two others that studied the NH and HI listeners separately. We estimated that there was not enough information to use informative priors and decided to use a Cauchy prior distribution as a default prior, with a scale value of 1/2 [37]. Indeed, as explained above, to the best of our knowledge there is no clear evidence to confirm or reject hypotheses 1, 1-1, 1-2, 2, 3-2-a and 3-2-b. There are some studies that seem to contradict hypothesis 3-1, but they could be misleading due to the use of percentage scores instead of RAUs scores.

The three ANOVAs related to the three reverberation conditions had one between-group factor (HI versus NH) and two within-group factors (RT and noise modulation). For each of these Bayesian ANOVAs, the Bayes factor comparing the models with the interaction studied (RT × group or RT × noise × group) and without this interaction was calculated. The two ANOVAs related to each listener group only had two within-group factors (RT and noise modulation). A Bayes factor superior to 10 was considered strong evidence for the existence of the model effect, while a Bayes factor inferior to 1/10 was considered strong evidence for its absence. A Bayes factor above 3 or below 1/3 was seen as a moderate evidence, while a Bayes factor between 3 and 1/3 did not allow for any conclusion [38].

Each of these ANOVAs had a different goal:

— The first one applied to the 'speech only' condition primarily tested the interaction RT × group, to determine whether NH and HI listeners are affected differently by the temporal smearing of speech caused by reverberation. This tested hypothesis 1.
— The second one applied to the 'noise only' condition, investigated the interaction RT × group × modulation type, to see whether HI and NH listeners are differently affected by the decrease of dip listening benefit caused by reverberation. This tested hypothesis 2.
— The third one applied to the 'speech + noise' condition looked at the interaction RT × group, to see whether HI and NH listeners are affected differently by reverberation when it is applied to both sources. This tested hypothesis 3-1.
— The fourth and fifth ones, applied to the 'speech + noise' condition looked at the interaction RT × noise for the NH and HI listeners separately, to determine for each group whether the dip listening decrease is significant when reverberation is applied to speech and noise. This tested hypotheses 3-2-a and 3-2-b.

To deepen our analyses, *post hoc* tests were also performed. To test hypothesis 1-1, Bayesian repeated measures *t*-tests compared the intelligibility scores between RT = 0.15 s and the other RTs, for each group of listeners and each noise modulation. This determined the lowest RT at which each group

was first affected by reverberation. To test hypothesis 1-2, Bayesian repeated measures *t*-tests compared the score difference between RT = 0.15 s and RT = 1.5 s across groups. This determined whether HI listeners are more affected than NH listeners at high RTs.

In addition to these tests, another ANOVA was applied across the three applications of reverberation ('speech only', 'noise only', 'speech + noise'). This test looked at the effects of hearing loss and the application of reverberation on intelligibility scores. Since these two effects were near certain, this ANOVA acted as a form of positive control and was not included in the analysis plan.

### 2.5.2. Sampling plan

With a Bayesian analysis, statistical power is not as much a problem as it is with a frequentist analysis. Indeed, since Bayesian tests can provide evidence for the null hypothesis and the alternative hypothesis, it is possible to accumulate participants until the Bayes factor reaches a critical value or until a preset maximum number of participants has been reached [39]. Here, it was decided to do a Bayes factor design analysis [40], more specifically a SBF + maxN design. The objective of these power analyses was to determine the probability of obtaining misleading evidence or inconclusive results. It was done with a Monte Carlo method [41].

Based on previous results, a Monte Carlo analysis was done only for the hypothesis 3-1. Data are lacking to make any accurate power analysis for the other hypotheses. The intelligibility scores in the various conditions were randomly generated from Gaussian distributions having the means and standard deviations measured by Harris & Swenson [16] in RAUs. The Bayesian ANOVA was then applied with the minimum number of listeners (16 NH and 16 HI listeners). The number of listeners was then raised progressively (two by two, one supplementary listener for each group), each time repeating the ANOVA until a Bayes factor superior to 3 or inferior to 1/3 was reached for the RT × group interaction, or until the maximum number of listeners (64 listeners) was reached. This procedure was repeated 2000 times.

It was estimated that, with a minimum of 32 listeners and a maximum of 64 listeners (in total, NH + HI), there was a probability of 0.82 to confirm the absence of a RT × group interaction, a probability of 0.16 to prove the existence of an interaction (contradicting our hypothesis) and a probability of 0.02 to reach the maximum number of listeners without being able to reach a conclusion. When the maximum number of listeners was reached, there was a probability of 0.47 that the Bayes factor pointed in the hypothesized direction (Bayes factor inferior to 1). The mean number of listeners necessary to reach a conclusive Bayes factor was estimated at 36.

Based on these analyses, it was decided to test a maximum of 64 listeners (32 for each group). First a minimum of 32 listeners was tested (16 per group). The data were then analysed. New listeners were then tested until the maximum number of listeners was reached, or until a Bayes factor above 3 or below 1/3 was reached for all the following tests:

— RT × group interaction in the 'speech only' condition ((hypothesis 1).
— RT × noise × group modulation in the 'noise only' condition (hypothesis 2).
— RT × group interaction in the 'speech + noise' condition (hypothesis 3-1).

The design table can be found in the electronic supplementary material, table S-1, in 4.

### 2.5.3. Additional analyses

Because the test for hypothesis 3-1 remained inconclusive, the maximum number of listeners (32 NH and 32 HI listeners) was tested. The planned analyses were done as announced, but some additional analyses were deemed necessary to further interpret the results.

For each application of reverberation (speech, noise, speech + noise), the tests were done at an 'optimal' SNR of the RAU conversion of the fitted psychometric curves. For comparisons between groups, this SNR was common to the HI and NH listeners. As the chance level was equal to 10% in this experiment, this SNR was chosen as the one with the maximum of intelligibility scores between 15% and 95%, in order to minimize floor and ceiling effects. However, the effect of dip listening and its interaction with the effect of reverberation varied a lot depending on the SNR considered. For some SNRs, these effects were significantly present for both groups, while they were completely absent for other SNRs. To do a more precise analysis, the tests studying these effects were also done at the SNR where the listeners had the maximum amount of dip listening benefit. This concerns the pre-planned tests 2, 3-1, 3-2-a and 3-2-b.

Two (control) Bayesian ANOVAs were also done across all conditions to confirm that the results varied depending on the application of reverberation (to apply reverberation only to the target speech, only to the noise masker, or to both) and the listener group. Both of these ANOVAs found significant effects at the optimal SNR (BF = $10^{109}$ for the application of reverberation, and BF = $5 \times 10^{106}$ for the listener group).

There were difficulties to recruit older NH listeners, and there is a 5-year difference between the HI and NH listeners. To make sure that there was no significant age effect on the results, the three tests comparing the monaural effects of reverberation between HI and NH groups (hypotheses 1, 2, 3-1) were applied to smaller HI and NH groups with matched age. These results can be found in electronic supplementary material, figures S-1 to S-3 and table S-2 in 4. The statistical tests were also applied to the estimated SRTs of the listeners, in order to provide the community with an alternative view of the same dataset to enable easy comparison with both previous and future experimental studies and models. The mean SRTs can be found in electronic supplementary material, figures S-4 to S-6, and the results can be found in electronic supplementary material, table S-3. Several positive controls were also added in the electronic supplementary material to confirm the presence of the studied effects. These results can be found in electronic supplementary material, table S-4. All the tests (planned and additional) were also done at all SNRs of the RAU conversion of the psychometric curves. These results can be found in electronic supplementary material, figures S-7 to S-16 and table S-5.

# 3. Results

For each application of reverberation, the planned and additional analyses are detailed separately below. A summary of the results can also be found in table 4.

## 3.1. Reverberation applied only to the speech target

### 3.1.1. Planned analyses

The RAU conversion of the mean psychometric curves of each group in the condition where reverberation was applied only to the target speech can be found in figure 4. For both groups, there was a strong effect of the temporal smearing of the speech. The planned analyses were done at the optimal SNR of −4.4 dB, where 99% of the results were within the floor and ceiling limits. At this optimal SNR, for the HI listeners, the mean score impairment due to the temporal smearing of speech (that is to say, the score difference between the RT = 0.15 s condition and the other conditions) increases from 17 RAU at RT = 0.5 s to 32 RAU at RT = 1.5 s. For the NH listeners it increases from 14 RAU at RT = 0.5 s to 33 RAU at RT = 1.5 s. The ANOVA studying hypothesis 1 found that the temporal smearing of the speech had the same effect for the HI and NH listeners (BF = 0.02). The t-tests done at each RT to study hypothesis 1-1 found a significant score impairment due to reverberation starting at RT = 0.5 s for each group (BF = $5 \times 10^6$ for the HI listeners and BF = $10^{11}$ for the NH listeners). The t-test done to compare the score differences of each group between RT = 0.15 s and RT = 1.5 s (hypothesis 1-2) found a significant absence of difference (BF = 0.26).

### 3.1.2. Additional analyses

In order to further compare the groups, additional t-tests were done. For each group the score differences between RT = 0.15 s and the other RTs were calculated. The score differences of the HI and NH listeners were then compared with Bayesian t-tests. At RT = 0.5 s, the test was inconclusive (BF = 1.23), while there was a similar impairment caused by reverberation between groups at RT = 0.8 s and RT = 1.1 s (BF = 0.22 and BF = 0.28, respectively).

## 3.2. Reverberation applied to the noise masker

### 3.2.1. Planned analyses

The RAU transformation of the mean psychometric curves of each group in the condition where reverberation was applied only to the noise masker can be found in figure 5. In this case, the planned analysis studying the difference of reduction in dip listening benefit between groups (hypothesis 2) was done at the SNR of −6.3 dB, where 100% of the results were within the floor and ceiling limits.

**Table 4.** Summary of the results of the planned and additional analyses.

| hypothesis | test | SNR | result | interpretation |
|---|---|---|---|---|
| 1: The temporal smearing of the speech caused by reverberation impairs more HI listeners than NH listeners. | ANOVA (RT × group interaction) | SNR = −4.4 dB; percentage of conditions without floor and ceiling effects: 99% (for both groups) | BF = 0.02 | At the optimal SNR the hypothesis is rejected. |
| 1-1: The temporal smearing of the target speech impairs intelligibility at lower levels of reverberation for HI listeners compared with NH listeners. | $t$-tests between RTs | | BF > 10 for both groups at all RTs | At the optimal SNR the hypothesis is rejected. |
| 1-2: At high levels of reverberation, HI listeners are more impaired than NH listeners by the temporal smearing of the speech. | $t$-test to compare the score difference of the groups between RT = 0.15 s and RT = 1.5 s | | BF = 0.26 | At the optimal SNR the hypothesis is rejected. |
| The magnitude of the effect of reverberation on speech is only higher for HI listeners at low RTs. | $t$-tests to compare the difference in intelligibility scores between RT = 0.15 s and the intermediary RTs for both groups | | At −4.4 dB: BF = 1.2 at RT = 0.5 s, BF = 0.2 at RT = 0.8 s, BF = 0.3 at RT = 1.1 s | At the optimal SNR no conclusion is possible at RT = 0.5 s. The hypothesis is rejected for RT = 0.8 s and 1.1 s. |
| 2: Reverberation affects the dip listening benefit more for NH listeners than for HI listeners. | ANOVA (RT × noise × group interaction) | SNR = −6.3 dB and −10.3 dB; percentage of conditions without floor and ceiling effects: 100% (for both groups) at −6.3 dB. 100% (for the NH listeners) and 92% (for the HI listeners) at −10.3 dB | At −6.3 dB, BF = 0.26; at −10.3 dB, BF = 3.13 | At the optimal SNR, the hypothesis is rejected. At the SNR with the maximum amount of dip listening benefit for the HI listeners, the hypothesis is accepted. |

**Table 4.** (*Continued.*)

| hypothesis | test | SNR | result | interpretation |
|---|---|---|---|---|
| There is a difference in dip listening benefit between groups at the lowest level of reverberation ($RT = 0.15$ s) | ANOVA (group × noise interaction) | | At −6.3 dB, BF = 0.2; at −10.3 dB, BF = 4.17 | At the optimal SNR the hypothesis is rejected, but at the SNR with the maximum dip listening benefit for HI listeners the hypothesis is accepted. |
| 3-1: The overall (monaural) effect of reverberation impairs HI listeners as much as NH listeners. | ANOVA (RT × group interaction) | SNR = −4.8 dB and −10.3 dB; percentage of conditions without floor and ceiling effects: 99% (for both groups) at −4.8 dB. 92% (for the NH listeners) and 81% (for the HI listeners) at −10.3 dB | At −4.8 dB, BF = 1.46; at −10.3 dB BF = $3 \times 10^4$ | At the optimal SNR no conclusion is possible. At the SNR with the maximum amount of dip listening benefit for the HI listeners, the hypothesis is accepted. |
| 3-2-a: In the case of NH listeners the dip listening decrease has a significant effect on intelligibility when reverberation is applied to speech and noise. | ANOVA (RT × noise interaction for the NH group) | SNR = −4.8 dB and −11.8 dB; percentage of conditions without floor and ceiling effects: 99% and 73% respectively | At −4.8 dB, BF = 0.76; at −11.8 dB, BF = $4 \times 10^{14}$ | At the optimal SNR, no conclusion is possible. At the SNR with the maximum amount of dip listening benefit for the NH listeners, the hypothesis is accepted. |
| 3-2-b: In the case of HI listeners the dip listening decrease has a significant effect on intelligibility when reverberation is applied to speech and noise. | ANOVA (RT × noise interaction for the HI group) | SNR = −4.8 dB and −10.3 dB; percentage of conditions without floor and ceiling effects: 99% and 81%, respectively | At −4.8 dB, BF = 0.46; at −10.3 dB, BF = 13.13 | At the optimal SNR, no conclusion is possible. At the SNR with the maximum amount of dip listening benefit for the HI listeners, the hypothesis is accepted. |

At this SNR, with the lowest reverberation time of 0.15 s, the dip listening benefit (that is to say, the score difference between the conditions with the modulated and stationary noises) was very limited for both groups, at 5 RAU for the HI listeners and 6 RAU for the NH listeners. At the highest reverberation time of 1.5 s, the listeners had slightly worse scores in the modulated noise condition, with a 2 RAU decrease and a 1 RAU decrease for the HI and NH listeners, respectively. At this SNR, the ANOVA testing for hypothesis 2 found no difference of reduction of dip listening between groups (BF = 0.26).

### 3.2.2. Additional analyses

The difference of reduction of dip listening benefit between groups was also studied at −10.3 dB, where HI listeners had the maximum amount of dip listening benefit (9 RAU) and the NH listeners were close to their maximum benefit (20 RAU at −10.3 dB against 21.5 RAU at −11.8 dB). At this SNR 100% of the results were within the floor and ceiling limits for the NH listeners, while only 92% were for the HI listeners.

At this SNR, with a reverberation time of 0.15 s, the dip listening benefit was 9 RAU for the HI listeners while it was 20 RAU for the NH listeners. With a reverberation time of 1.5 s, the dip listening benefit was reduced to 1 RAU for both groups. At this SNR, there was a significant RT × noise type × group interaction (BF = 3.1), indicating that the reduction of dip listening benefit by reverberation was significantly larger for NH listeners compared with HI listeners.

One additional test was also added as positive control. The dip listening benefit was compared between HI and NH listeners with an ANOVA. This test was done at the SNRs −6.3 and −10.3 dB. A significant lack of interaction group × noise type was found at −6.3 dB (BF = 0.2), while a significant interaction was found at −10.3 dB (BF = 4.2), indicating that the dip listening benefit was then larger for NH than HI listeners.

## 3.3. Reverberation applied to both speech and noise

### 3.3.1. Planned analyses

The RAU transformation of the mean psychometric curves of each group in the condition where reverberation was applied only to both the speech target and the noise masker can be found in figure 6. The planned analysis studying the difference of effect of reverberation between groups (hypothesis 3-1) was done at the optimal SNR −4.8 dB, where 99% of the results were within the floor and ceiling limits. At this SNR, the score difference between the conditions with the maximum and minimum RTs was 38 RAU for the NH listeners and 32 RAU for the HI listeners. The test was inconclusive at this SNR (BF = 1.46).

The planned analyses studying the dip listening reduction caused by reverberation for the NH listeners (hypothesis 3-2-a) and separately for the HI listeners (hypothesis 3-2-b) were also done at −4.8 dB. At this SNR the tests were found inconclusive for both groups (BF = 0.76 and BF = 0.46 for the NH and HI listeners, respectively).

### 3.3.2. Additional analyses

Since the planned test 3-1 was inconclusive at the optimal SNR, it was also done at the SNR where the HI listeners had their maximum amount of dip listening benefit. As the condition with the lowest RT is the same for the three applications of reverberation, this SNR was −10.3 dB, as before. In this case, 92% of the results were within the floor and ceiling limits for the NH listeners against 81% for the HI listeners. The score difference between the conditions with the maximum and minimum RTs was 17 RAU for the NH listeners and 11 RAU for the HI listeners. The global monaural effect of reverberation was found significantly different between groups at this SNR (BF = 31 390).

The tests planned for hypotheses 3-2-a and 3-2-b were also done at the SNR where each group had the maximum amount of dip listening benefit: −10.3 dB for the HI listeners, −11.8 dB for the NH listeners. In this case 81% of the results were within the floor and ceiling limits for the HI listeners, while only 73% were for the NH listeners. At these SNRs, there was a significant dip listening reduction caused by reverberation for both groups (BF = $4 \times 10^{14}$ and BF = 13 for the NH and HI listeners respectively).

A summary of the results (planned and additional) can be found in table 4.

# 4. Discussion

The aim of this study was to compare two monaural effects of reverberation on speech intelligibility for HI and NH listeners. When comparing intelligibility performances, it appears that the choice of the SNR at which the comparison is done can greatly affect the results. At the optimal SNR limiting floor and ceiling effects, reverberation had a strong influence on intelligibility scores through its temporal smearing of the target speech, but very little through its temporal smearing of the noise (affecting dip listening). The tests at the SNR maximizing the dip listening benefit for the HI listeners allowed to study the largest effect that reverberation can have on this dip listening benefit, even if individual floor performances can then influence the mean results of each group.

When reverberation was applied only to the target speech, there was an important score reduction for both groups, and there was a significant lack of difference of this effect of temporal smearing of the speech between groups at the optimal SNR. The results also showed a significant effect of reverberation for both groups at all RTs. There was a significant lack of difference of this effect of reverberation between groups at all RTs except 0.5 s (where no conclusion is possible). This indicates that the effect of the temporal smearing of the target is similar for the two groups at least at intermediary and high RTs. Overall, we conclude that the temporal smearing of the speech had a similar effect for HI and NH listeners, even if low reverberation levels (below 0.8 s) could be further investigated.

When reverberation was applied only to the noise masker, the dip listening benefit was completely cancelled for both groups. The test for hypothesis 2 showed that there was a significant difference of this effect of reverberation on the noise between groups at −10.3 dB, the SNR where the HI listeners had their maximum amount of dip listening benefit. This interaction could highlight a difference of effect of reverberation on the dip listening benefit between groups. However, it could also be due to a lower dip listening benefit for the HI listeners to start with, in the almost anechoic condition (RT = 0.15 s). The results indeed indicate that the dip listening benefit is significantly smaller for the HI listeners at RT = 0.15 s. So the smaller reduction of dip listening benefit caused by reverberation for these listeners could be solely due to their reduced ability to benefit from the masker dips to start with, even in the absence of reverberation, as also observed by Jensen & Bernstein [24]. The observed interaction could also be partly influenced by floor effects in the individual intelligibility scores of the HI listeners. Overall, we cannot conclude here that the reduction of dip listening benefit caused by reverberation was more detrimental for HI listeners than for NH listeners.

When reverberation was applied to both speech and noise, there was an important score reduction for both groups. This effect of reverberation was significantly different between HI and NH listeners only at the SNR maximizing dip listening. The test was inconclusive at the optimal SNR, even if both individual effects of reverberation were found similar for the two groups at this particular SNR, when studied in isolation. It is interesting to note that, in this condition, the dip listening reduction caused by reverberation was found significantly present for both groups at the SNR maximizing dip listening, but not at the optimal SNR. This could indicate that most of the difference between groups at the SNR maximizing dip listening comes from the effect on the dip listening benefit, which cannot be directly associated with the effect of reverberation, as discussed in the previous paragraph. The score differences between conditions also indicate that the dip listening reduction caused by reverberation has a less important effect on speech intelligibility than the temporal smearing of the speech. Again, we cannot conclude that reverberation was more detrimental for the HI listeners in this condition (reverberation applied on both speech and noise).

There was a significant age difference between the listener groups. Nábělek & Robinson [42] compared the effect of temporal smearing of the speech by reverberation for NH listeners of mean ages going from 10 to 72 years old. Their two older groups (mean ages of 64 and 72 years old) did not have a significant intelligibility score difference [42]. Here, the NH group had a mean age of 62 years old, while it was 67 years old for the HI listeners. This age difference is less than the difference between the two older groups that had similar intelligibility scores in the study of Nábělek and Robinson. We can thus assume that the age difference in the present study was too small to significantly affect its results. In a similar way, while the dip listening benefit can vary depending on the age of the listeners [43], differences are usually found between groups with large age differences. To confirm the absence of a significant age effect, supplementary analyses were done with age-matched groups in the electronic supplementary material. These analyses lead to results very close to the results from the main analyses, such that an effect of age is rather unlikely in our data.

The SRT analysis provided in the electronic supplementary material indicates an effect of the temporal smearing of the target for both groups, but only finds a dip listening benefit for the NH listeners. This is most likely due to the confounding effect that the SNR has when evaluating the dip listening benefit [24]. This analysis did not find any conclusive results for the comparisons between HI and NH listeners, highlighting the importance of the analysis of the full psychometric functions done in the present study. The tests done at each SNR of the psychometric curves, also provided in the electronic supplementary material, permitted a broader view of the effects studied. As expected, for very low (respectively, very high) SNRs, both groups are at floor (respectively, at ceiling) and the results are significantly similar across groups. For intermediate SNRs, a single group among the two is either at floor or at ceiling and the results evidence a significant difference across groups. This is the case for the effects of the temporal smearing of the speech at high SNRs and the decrease of dip listening benefit at low SNRs. This difference is then mainly due to the floor or ceiling effects. Finally, the effect of the other factors (i.e. reverberation, noise modulation, group of listeners) can only be interpreted when all floor and ceiling effects across listeners are avoided (i.e. around the optimum SNR). As such, the presence of floor and ceiling effects in the individual results strongly affected the results and must be taken into account. Some previous results from the literature might have to be re-interpreted taking this into account.

The present study could be extended to consider binaural configurations. In these configurations, there are two additional mechanisms that affect intelligibility and can be influenced by reverberation: better-ear listening and binaural unmasking. Better-ear listening relies on the fact that the listener can use the information coming from the ear providing the better SNR (at a given moment, in a given frequency band). Binaural unmasking is the release from masking provided by a difference in interaural time difference (ITD) between the target and masker signals that allows the central auditory system to cancel part of the masker [44]. Several studies have shown that these mechanisms are impaired by reverberation [45–47]. The sound reflections travelling around the listener reduce the SNR difference between the ears, impairing better-ear listening [45]. Moreover, because the reflections are not identical at the two ears, they decorrelate the interfering sound at the two ears, reducing the efficiency of binaural unmasking [26]. Vicente *et al.* [48] compared the effect of reverberation on binaural unmasking for HI and NH listeners and found very similar results between groups, with only a 0.6 dB difference of mean SRT. The effect of reverberation on better-ear listening could be linked to the monaural effect that reverberation has on dip listening. Since it has been shown here that this monaural effect was similar for HI and NH listeners, it is possible to make the hypothesis that the effect of reverberation on better-ear listening will not be much different between groups. In the end, even if this certainly deserves further investigation, the overall deleterious effect of reverberation (all monaural and binaural effects considered) seems to be very comparable for NH and HI listeners.

## 5. Conclusion

Intelligibility scores were compared for NH and HI listeners, in the presence of stationary and modulated noises, at different levels of reverberation, which was applied only on the speech source, only on the noise source, or on both sources simultaneously. Most of the differences in the effects of reverberation between NH and HI listeners were not observed at the optimal SNRs limiting floor and ceiling effects in the individual data, but at the SNR maximizing dip listening, and might be explained by factors other than reverberation. The temporal smearing of the speech had a similar effect for the HI and NH listeners at the optimal SNR. For both groups the dip listening benefit was very limited at the optimal SNRs, so that the temporal smearing of the noise by reverberation had the same effect for the HI and NH listeners too. At the SNR maximizing dip listening, the dip listening benefit and the reduction of dip listening benefit caused by reverberation were more important for the NH listeners. The temporal smearing of the noise was thus less detrimental for the HI than for the NH listeners, but this was most likely due to the fact that HI listeners were experiencing less dip listening advantage to start with (in the absence of reverberation) rather than to a different effect of reverberation between groups. The test comparing the combination of the two monaural effects of reverberation for HI and NH listeners was inconclusive at the optimal SNR, even if, as previously mentioned, there were no differences between the listener groups for each of the two individual effects at this SNR. There was a significant difference at the SNR maximizing dip listening, but again it could be due to a reduced ability to benefit from dip listening for HI listeners (rather than to an effect of reverberation), as well as to floor effects in the individual HI data. Overall, when controlling for several biases such as the

confounding effect of SNR, floor and ceiling effects, and the age of the listeners, reverberation was not found more detrimental for the HI listeners in the present study.

Ethics. The study has received a formal agreement from a French national ethic committee (CPP Ile de France X, protocol no. 05-2019).

Data accessibility. The approved stage 1 protocol is accessible with the following link: https://zenodo.org/record/5031294#.YhTkVujMJPa. All data, test programs and data analysis scripts were provided on the Zenodo full open source platform and are accessible with the following link: https://zenodo.org/record/6782694#.Yr1tEnZByUk. The speech material is under copyright from the company Hörtech gGmbH and cannot be provided. Electronic supplementary materials were archived in Zenodo and are accessible with the following link: https://zenodo.org/record/6542359#.YnzkJ-hByUk and electronic supplementary material is available online [49].

Authors' contributions. R.C.: conceptualization, data curation, formal analysis, investigation, methodology, project administration, validation, visualization, writing—original draft, writing—review and editing; M.L.: conceptualization, funding acquisition, methodology, project administration, supervision, validation, writing—review and editing; N.G.: conceptualization, funding acquisition, methodology, resources, supervision, validation, writing—review and editing.

All authors gave final approval for publication and agreed to be held accountable for the work performed therein.

Conflict of interest declaration. We declare that we have no competing interests.

Funding. This research has been conducted within the framework of the Labex CeLyA (grant no. ANR-10-LABX-0060).

Acknowledgements. We acknowledge Maxime Bonneuil, audioprothesist, for his precious help to recruit hearing-impaired participants with particular audiometric specifications. We would like to thank the anonymous reviewers who took the time to make very constructive remarks that greatly improved this study. We would also like to thank the HörTech gGmbH company who allowed us to use the copyright-protected audio material of the French Matrix Test.

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
