## [Peer Review File · Royal Society Open Science]

Review History

Decision letter (RSOS-202302.R0)

Dear Mr Cueille,

I write you in regards to manuscript RSOS-202302 entitled "Effects of reverberation on speech intelligibility in noise for hearing-impaired listeners" which you submitted to Royal Society Open Science.

We routinely triage submissions for scientific soundness, clarity and general adherence to the Registered Reports guidelines. For submissions that have promise but are not yet suitable for in-depth Stage 1 review, we offer feedback to help authors maximise the chances that reviewers will respond positively to a resubmission.

We have concluded that your submission is not yet suitable for in-depth review and has therefore been rejected at this time, but we believe it will be suitable once several issues are addressed. We therefore invite a resubmission. Further comments from the Associate Editor may be found at the end of this letter.

If you wish to revise your manuscript in light of the below comments please submit your manuscript as a new submission and mention this previous manuscript ID in your covering letter. You should also provide a detailed response to the below comments in the cover letter.

Thank you for considering Royal Society Open Science for the publication of your registered report.

on behalf of Professor Chris Chambers (Registered Reports Editor, Royal Society Open Science)
openscience@royalsociety.org

Associate Editor Comments to Author:

Associate Editor

Comments to the Author:

1. Please ensure that there is a direct correspondence between the proposed hypotheses, the sampling plans, and the proposed statistical tests in every case. At present these links are somewhat present but are limited to prose descriptions and too vague. To make the review process as straightforward as possible, please include a design table for the proposed study in the Method section of the main text based on Section 9 of this template: <https://osf.io/93znh/> You can find published examples of how these appear in these submissions (from different fields, but the principles are the same): <https://osf.io/xhdpu>, <https://osf.io/ym8gc>, <https://osf.io/g9dxb/>. Please ensure that each prediction is associated with a statistical sampling plan, a specific test (or set of tests) on specifically defined variables, and a comprehensive interpretative plan given different outcomes (i.e. a precommitment to different conclusions given different results).

2. Related to point 1, please consider whether the prespecified analyses themselves are sufficiently precise to confirm or disconfirm the hypotheses. For some predictions the authors predict interactions, but interactions can be supported multiple different datapatterns, not all of which may be consistent with the respective prediction. For example: "...whether HI and NH listeners are affected differently by reverberation when it is applied on both sources." Is any differential effect sufficient confirmatory evidence? Is the key confirmatory support for the predictions, in the end, supported by specific pairwise differences between conditions, or combinations thereof? The authors mention post hoc tests, but it is not clear whether these are necessary to confirm/disconfirm the hypotheses or are more exploratory.

3. I find the sampling plan to be a slightly strange mix of Bayesian and frequentist methods. Are the authors employing a Bayes Factor Design Analysis (BFDA; see <https://link.springer.com/article/10.3758/s13423-017-1230-y>)? This may be a more coherent approach. I suspect the current method will draw criticism from statistical reviewers. Please also make clear what prior will be used in the Bayesian analyses and why. Minor point: "bayesian" should be capitalised throughout, and it is "Bayes factor", not "bayesian factor"

4. One of the key criteria that reviewers are asked to assess in Stage 1 RRs is “Whether the authors have considered sufficient outcome-neutral conditions (e.g. absence of floor or ceiling effects; positive controls; other quality checks or sanity checks) for ensuring that the results obtained are able to test the stated hypotheses”. Such tests should be orthogonal of the main hypotheses. Your study does not obviously propose such tests (e.g. confirming, independent of the main outcomes, that the data quality was sufficient to be diagnostic about the hypotheses). Please consider whether such positive controls or sanity checks are necessary, and if possible how they might be included in the design.

5. Authors usually prepare their Stage 1 RR submissions in future tense to avoid reviewers from concluding (mistakenly) that the work has already been done with the results merely redacted. If the authors prefer past tense (which does have the advantage of efficiency at Stage 2), then I stringly suggest adding a cover note to the top of the manuscript making it clear to reviewers that although the rationale and methods are for expediency written in past tense, none of the research has yet been undertaken.

Author's Response to Decision Letter for (RSOS-202302.R0)

See Appendix A.

RSOS-210342.R0

Review form: Reviewer 1

Do you have any ethical concerns with this paper?

No

Recommendation?

Accept with minor revision

Comments to the Author(s)

General comments.

The proposed expeeriment examines the interacting effects of masker modulation and room reverberation on speech intelligibility for hearing-impaired (HI) listeners (compared to normally hearing (NH) listeners). The design and methods of analysis look well-suited to answering the set of proposed hypotheses. The authors have negotiated some difficult design problems that lie unaddressed in most previous work in this area. These problems are described in the Introduction to some extent, but perhaps not as directly as they might be. One problem with many of the studies descibed is that those studies report percentage changes in word identification, but the relative sizes of such changes are often dependent upon the baseline level of intelligibility (where modulation and/or reverberation are absent). Since the baseline is different for NH and HI listeners, intelligibility is nearer floor or nearer ceiling in comparison conditions, making the changes in percentage difficult to interpret. Many studies avoid floor and ceiling effects by measuring shifts in speech reception threshold, but these studies consequently compare NH and HI performance at different signal-to-noise ratios. One study that they describe

(Jepsen and Bernstein) elegantly skirted these problems by first equalising NH and HI intelligibility through variations in the size of the response set. The present study aims to achieve the same end by transforming percent correct scores at the same signal-to-noise ratios into rationalised arcsine units which reduce the influence of floor and ceiling effects. I have to say that I do prefer the Jepsen and Bernstein approach, and I wonder whether it might be possible to adopt it in the present study. It might be straightforward to change the number of different tokens used in the frMatrix test so that NH and HI performance is equalised, but it would require some pretesting to calibrate the required numbers.

Beyond this, the study will examine for the first time the relative influences of reverberation on the target and interfering noise in HI listeners, as well as the interaction with masker modulation. This all looks like an excellent plan.

Minor points (page #s from top left corner).

P2, L25. This may need rephrasing. There are also monaural effects on the processing of F0 differences.

P4, L10 and elsewhere. It is commonplace to specify reverberation by the reverberation time, but it is, from the point of view of the effect on temporal smearing, a very tangential parameter. The important factor is the direct-to-reverberant ratio. For a fixed room and source/receiver geometry (as used in the proposed expt.) direct-to-reverberant ratio and reverberation time will vary in the same way, so there is internal consistency. However, comparisons across experiments, which may use different room sizes and/or source distances, become invalid unless transformed into direct-to-reverberant ratio using details of the geometry. It would be good to at least adopt the habit of reporting both RT and D/R in order to facilitate such comparisons. On this particular line, RT=0.25 s and RT=0.5 s are described as "low", but I cannot tell from these numbers alone whether the D/R was low.

P4, L39-41. The Jepsen and Bernstein method could be described in more detail here.

P5. I found it hard to digest the hypotheses before the following paragraph summarising the method.

I was a bit puzzled by hypotheses 4,4-1,4-2, since 1, 2 and 3 looked as though they were contingent on the same things. Maybe these are fall-back hypotheses.

P7. Again, it would be nice to see the D/Rs for these conditions.

P7, last para. I was quite confused at this point, because the use of rationalised arc-sine units had not yet been mentioned, and I didn't see how these things could be legitimately taken from the data.

Review form: Reviewer 2

Do you have any ethical concerns with this paper?

No

Recommendation?

Accept with minor revision

Comments to the Author(s)

This is a carefully designed study that ought to add knowledge to the question of the effect of reverberation on hearing-impairment. Its encouraging for the future of our field that a PhD student wishes to write a registered report – well done

However certain choices need further exposition. Now, none of them are technically incorrect (excepting possibly the number-of-trials and the assumed priors), but the questions are raised to encourage reflection on each, as there may be better choices for each one that will improve the rigour and potential importance of the research:

- Why a room of 5m x 9m x 3 m, and why these particular positions within it (which reduces to the question of will this room only give results of particular interest to rooms just like this, or does it give generalizable results of much more interest?)
- Why only 10 trials per SNR? This really worries me, I'd have used 20 at least if not 30. Ten really is very low, and I am not convinced that it will give data of sufficient reliability to be able to properly answer the questions at hand.
- Why a presentation level of just 60 dB? That's fairly quiet I would have thought.
- Why don't do $rt = 0.2$ and 0.8 in the Noise & Speech-Noise conditions (table 3). I would have included them to complete the cells in the experiment.
- Why chose these particular RTs? Has a quantitative analysis of the modulations been done, that (ideally) demonstrates a reduction from high to low, with nice even steps inbetween?
- Is it fair to use the same prior distribution for all tests? To my mind from reading the introduction, hypothesis 1 is "fairly sure", #2 is "probable, but no actual evidence", #3 is "fairly sure" and #4 (see above) is "near-certain". Ought this difference in prior knowledge be reflected in the prior assumption for each test?
- In the analysis, what happened to the SNRs? There is no mention of "SNR" anywhere in the analysis section. Are they all being analyzed as individual points in the ANOVAs, or is a psychometric function to be fitted then that enters the ANOVA?
- Page 9 line 48 – does the Bayes criterion need be reached for all tests, or just one of them?
- Overall, why use a ANOVA in the first place? Both RTs and SNRs are on interval scales, so regression lines can be fitted to them, but ANOVAs don't care about such ordering (if A-B-C vs D-E-F is significant, so is C-B-A vs F-E-D etc). Why wasn't some form of regression analysis chosen?

Also, to be honest, 9 hypotheses is too many. It would be preferable to divide them into four primary hypotheses and the remainder as secondary. More importantly, hypothesis 4 – "hearing loss has an effect on speech intelligibility and the application of reverberation has an effect on speech intelligibility" – is surely entirely superfluous. I expect that there will not be any doubt in the field that the answers to both are Yes, so including them as hypotheses of the same status as the others dilutes the worth of the rest, and is certainly not of the level of scientific interest expected in a Royal Society journal. To my mind, it can be dealt with in a sentence in the experiment's results ("as was expected, HI and reverb affected speech intelligibility") before quickly moving onto results that do matter.

And for the record to answer the questions set

The scientific validity of the research question(s)

- Yes

-

The logic, rationale, and plausibility of the proposed hypotheses

- mostly yes, apart for hypothesis 4 and that nine in total is too many

-

The soundness and feasibility of the methodology and analysis pipeline (including statistical power analysis where applicable)

- mostly yes, apart from the comments above about the method & analysis.

-

Whether the clarity and degree of methodological detail would be sufficient to replicate exactly the proposed experimental procedures and analysis pipeline

- yes, apart from the issue of what happened to the SNRs in the analysis

-

Whether the authors provide a sufficiently clear and detailed description of the methods to prevent undisclosed flexibility in the experimental procedures or analysis pipeline

- mostly yes. To be honest, subsequent post-hoc analyses could always be done, but the primary ones as specified are clear enough to be followed.

-

Whether the authors have considered sufficient outcome-neutral conditions (e.g. positive controls) for ensuring that the results obtained are able to test the stated hypotheses

- I agree with the authors here, there's no need for a positive control on this.

PS Its Plomp not Plomb, Butterworth not butterworth, and "and" not "et"

Review form: Reviewer 3

Do you have any ethical concerns with this paper?

No

Recommendation?

Major revision

Comments to the Author(s)

Review of registred report – stage 1 “Effects of reverberation on speech intelligibility in noise for hearing-impaired listeners”

The statistical analysis planned in this study appears extremely elaborated to me and I highly appreciate the effort that has already been invested by the authors and the editor. I learned a lot and will try to approximate to these standards at least a little in my research. I cannot really contribute to the statistical aspects of this study.

I think, however, that I can contribute to the audiological aspects of this study.

The research questions and the list of hypotheses are without any doubt very interesting and the method appears in principle adequate to answer these questions. My main concern, however, is that I do not think that the stimuli planned to be used in this study are adequate. The original FrMatrix test has been realized very elaborately so that it preserves quite natural prosody and coarticulation between words as far as this is possible for the Matrix sentences concept and the

state of the art. Furthermore, the cautious selection of recordings and homogenization of the levels of the single words guarantees equal intelligibility across words, sentences and test lists. See for example the review of Kollmeier et al 2015 (DOI 10.3109/14992027.2015.1020971). The method proposed here, using rerecorded words, is similar to the first Matrix sentence test by Hagerman (1982, DOI 10.3109/01050398209076203) which was innovative for that time. Hagerman also recorded the words in isolation, but he did not adjust the levels for equal RMS. However, it is essential to understand that it is not an issue that the FrMatrix test has 280 sentences which is a smaller number as the total number required per listener in this study. It is virtually impossible to memorize these 280 sentences as they are all drawn from the same pool of 50 words and as they are always presented in random order. In fact none of the listeners will recognize that sentences are repeated and that most of the theoretically possible sentences never occur.

Nevertheless, there will be a learning issue in this study, but this learning effect is not due to the fact that the same sentences are used four or five times but due to the fact that each word will be used 112 to 140 times. This will cause a long term learning effect that cannot be avoided using the two 20 sentence lists for familiarization as recommended by Jansen et al. 2012 and as planned in this study. The initial familiarization reduces the learning effect by approximately 1 to 2 dB (see Hagerman 1982, Jansen et al 2012, and Kollmeier 2015). But the remaining learning effect of about one additional dB until 280 sentences is, for instance, reported by Hagerman and also by other authors. Unfortunately the learning effect continues even for longer measurements and across session as reported by Schlueter et al (2016, DOI 10.1177/2331216516669889). This long term-learning can be explained by the fact that the listeners develop to expert listeners for the words of the matrix and that they perform key word spotting (which is the best strategy) rather than real sentence recognition. The authors' method to address this problem by rerecording the sentences does not decrease this learning effect, but it may increase it by facilitating the word spotting.

This long-term familiarization is compensated to a certain degree by randomization of the test conditions but nevertheless it will introduce a lot of noise into the data. I therefore recommend compensating the long-term learning effect on an average basis. An even better solution would be not to use Matrix sentences at all, as simple daily life sentences do not show a training effect due to word spotting, but of course I see the problem that such test material with equal intelligibility across lists does not exist.

The rerecording of the words with neutral reflection is nonsensical and makes the problem that should be avoided even worse. I am sorry, that I have to express it so directly. The new construction of the recordings destroys prosody and coarticulation which are important cues for the building of acoustic streams and speech context which are main factors for dip listening. As dip listening is in the main focus of this study this has to be treated very cautiously.

Scaling the individual words to the same RMS might perhaps be convincing from a naïve technical point of view. But it is nonsensical from a speech processing point of view – sorry again. As different words are produced very differently they naturally differ in RMS as the averaging time is too short for a reliable and valid estimate of speaking effort. Instead it is recommended to record the words within sentences with cautious control of speaking effort by the talker and optionally adjust the levels of the sentences. Even better is the method applied in the matrix test concept like in the FrMatrix test, as it also adjusts the intelligibility between words improving the accuracy of each measurement. Therefore I strongly recommend to build on the experience of the quite high number of studies using the Matrix test concept and to use the original version of the FrMatrix test which is certainly better suited for this study than the material described in this manuscript.

The generation of the modulated noise is quite unnatural because such signals that are modulated like the broadband envelope of speech identically in all frequency bands hardly ever occur in real life. Speech is modulated slightly differently in different frequency bands. In order to increase ecological validity I recommend to use the modulated ICRA noises which show more speech-like modulations or to generate the modulated noise in a similar way as the ICRA noises (Dreschler et al, 2001).

The virtual room acoustics is very unnatural. Room with identical absorption for walls ceiling and floor do not exist. This contradicts the high effort for generating the rooms using CATT. This is not a severe problem for monaural measurements. But I could imagine that the authors want to extend their focus on binaural measurements. For such measurements the RIR would be really problematic.

It might be an issue that for each test condition 10 sentences containing the same pool of words are used. This depends on the manner of randomization mentioned below. This would have the consequence, that the later sentences in each block are biased by the words perceived in the previous trials as the best strategy is to repeat only words the listener hasn't heard so far in this list. In this case it is not a sentence intelligibility test but as a word spotting test combined with a short term memory span test. This is has not sufficient ecological validity to investigate the effects of reverberation and dip listening on sentence recognition.

The selection of the five fixed SNRs for measurements has not been specified clearly enough. In order to guarantee that NH and HI can be tested at the same SNRs, the intelligibility function of the test has to be known. As it is new material the intelligibility function is unknown, so far. As the single words have not been adjusted for equal intelligibility but for equal RMS, the slope of the intelligibility function can be expected to be flatter than in the FrMatrix test, see Kollmeier 2015 where the relation between slope and intelligibility of single words is explained. In order to specify the SNRs for testing the intelligibility function or at least the expected range has to be specified. I see that pilot measurements will be done, but it should be made clear which target intelligibilities should be met and if it is possible to fulfill the requirements of overlapping intelligibility ranges between conditions and listener groups.

P8 L7-8: what does it mean that the test condition is tested on a trial-by-trial order? Is a trial here a 10 sentence-list or a single sentence. I hope that the latter is meant. Then the memory span issue mentioned above is not an issue. But please specify how the tests are divided into blocks.

I also want to mention that there is evidence supporting the hypothesis that matrix sentences are less affected by reverberation than daily life speech (see for example Rennie et al 2014, 10.1121/1.4897398). More data are required concerning this question, but I think that this point should be discussed in this study.

Minor comments:

P1 L52: white space in "the"

References are in French style, e.g. "Helfer et Wilber (1990)".

P2 L39-58: due to the limitation of the scale and the non-linear intelligibility function it is problematic to compare intelligibility scores in %. If an intelligibility score decreases, for instance, from 100 to 80% it is a much stronger effect compared to a decrease from 60 to 40%. Later it is mentioned that RAUs will be used to compensate for this. It might be required to take this into account already here.

This also plays a role in P3 L12-13.

P41 L40: Why has the dip listening benefit to be evaluated at comparable SNRs? This is only one aspect. You can also argue that the dip listening benefit has to be sufficient high sensation level in both groups in order to assure that audibility is not the limiting factor in the dips, especially for the HI listeners.

P5 L53: Why do you plan to select the ear with the flatter hearing loss? One can also argue for selecting the better ear in order to reduce the effect of sensory deprivation due to the hearing loss of the weak ear.

In Beutelmann, Brand and Kollmeier (2010) different reverberation times and noises with different degrees of modulation have been tested with NH and HI listeners. The analysis is not as systematic as the analysis planned here, but there is at least one other study that investigated dip listening in reverberation in HI listeners.

P11: I think there is a typo in lower right field, second condition: "If $BF < 1/3$: the magnitude ..."

Decision letter (RSOS-210342.R0)

Dear Mr Cueille,

The Editors assigned to your Stage 1 Registered Report ("Effects of reverberation on speech intelligibility in noise for hearing-impaired listeners") have now received comments from reviewers. We would like you to revise your paper in accordance with the referee and editors suggestions which can be found below (not including confidential reports to the Editor). Please note this decision does not guarantee eventual acceptance.

Please submit a copy of your revised paper within three weeks (i.e. by the 26-May-2021). Please note that the revision deadline will expire at 00.00am on this date.

Kind regards,

Royal Society Open Science Editorial Office
Royal Society Open Science
openscience@royalsociety.org

on behalf of Professor Chris Chambers (Registered Reports Editor, Royal Society Open Science)
openscience@royalsociety.org

Associate Editor Comments to Author (Professor Chris Chambers):

Associate Editor: 1

Comments to the Author:

Three expert reviewers have now assessed the manuscript. All are positive and are generally very encouraging about the proposal, although there are some key issues to address to achieving Stage 1 in-principle acceptance. Broadly, the reviewers ask for clarification of the rationale in various places, stronger justification of design decisions, and additional clarity concerning procedures and analysis plans. More specifically, some headline issues to address include the adequacy of the stimuli (and concerns about long-term learning/familiarisation), clarity and number of the hypotheses, and trial numbers (along with a range of additional points). A major revision is invited to address these concerns.

Comments to Author:

Reviewer: 1

Comments to the Author(s)

General comments.

The proposed experiment examines the interacting effects of masker modulation and room reverberation on speech intelligibility for hearing-impaired (HI) listeners (compared to normally hearing (NH) listeners). The design and methods of analysis look well-suited to answering the set of proposed hypotheses. The authors have negotiated some difficult design problems that lie unaddressed in most previous work in this area. These problems are described in the Introduction to some extent, but perhaps not as directly as they might be. One problem with many of the studies described is that those studies report percentage changes in word identification, but the relative sizes of such changes are often dependent upon the baseline level of intelligibility (where modulation and/or reverberation are absent). Since the baseline is different for NH and HI listeners, intelligibility is nearer floor or nearer ceiling in comparison conditions, making the changes in percentage difficult to interpret. Many studies avoid floor and ceiling effects by measuring shifts in speech reception threshold, but these studies consequently compare NH and HI performance at different signal-to-noise ratios. One study that they describe (Jepsen and Bernstein) elegantly skirted these problems by first equalising NH and HI intelligibility through variations in the size of the response set. The present study aims to achieve the same end by transforming percent correct scores at the same signal-to-noise ratios into rationalised arcsine units which reduce the influence of floor and ceiling effects. I have to say that I do prefer the Jepsen and Bernstein approach, and I wonder whether it might be possible to adopt it in the present study. It might be straightforward to change the number of different tokens used in the frMatrix test so that NH and HI performance is equalised, but it would require some pretesting to calibrate the required numbers.

Beyond this, the study will examine for the first time the relative influences of reverberation on the target and interfering noise in HI listeners, as well as the interaction with masker modulation. This all looks like an excellent plan.

Minor points (page #s from top left corner).

P2, L25. This may need rephrasing. There are also monaural effects on the processing of F0 differences.

P4, L10 and elsewhere. It is commonplace to specify reverberation by the reverberation time, but it is, from the point of view of the effect on temporal smearing, a very tangential parameter. The important factor is the direct-to-reverberant ratio. For a fixed room and source/receiver geometry (as used in the proposed expt.) direct-to-reverberant ratio and reverberation time will vary in the same way, so there is internal consistency. However, comparisons across experiments, which may use different room sizes and/or source distances, become invalid unless transformed into direct-to-reverberant ratio using details of the geometry. It would be good to at least adopt the habit of reporting both RT and D/R in order to facilitate such comparisons. On this particular line, RT=0.25 s and RT=0.5 s are described as "low", but I cannot tell from these numbers alone whether the D/R was low.

P4, L39-41. The Jepsen and Bernstein method could be described in more detail here.

P5. I found it hard to digest the hypotheses before the following paragraph summarising the method.

I was a bit puzzled by hypotheses 4,4-1,4-2, since 1, 2 and 3 looked as though they were contingent on the same things. Maybe these are fall-back hypotheses.

P7. Again, it would be nice to see the D/Rs for these conditions.

P7, last para. I was quite confused at this point, because the use of rationalised arc-sine units had not yet been mentioned, and I didn't see how these things could be legitimately taken from the data.

Reviewer: 2

Comments to the Author(s)

This is a carefully designed study that ought to add knowledge to the question of the effect of reverberation on hearing-impairment. Its encouraging for the future of our field that a PhD student wishes to write a registered report – well done

However certain choices need further exposition. Now, none of them are technically incorrect (excepting possibly the number-of-trials and the assumed priors), but the questions are raised to encourage reflection on each, as there may be better choices for each one that will improve the rigour and potential importance of the research:

- Why a room of 5m x 9m x 3 m, and why these particular positions within it (which reduces to the question of will this room only give results of particular interest to rooms just like this, or does it give generalizable results of much more interest?)
- Why only 10 trials per SNR? This really worries me, I'd have used 20 at least if not 30. Ten really is very low, and I am not convinced that it will give data of sufficient reliability to be able to properly answer the questions at hand.
- Why a presentation level of just 60 dB? That's fairly quiet I would have thought.
- Why don't do $rt = 0.2$ and 0.8 in the Noise & Speech-Noise conditions (table 3). I would have included them to complete the cells in the experiment.

- Why chose these particular RTs? Has a quantitative analysis of the modulations been done, that (ideally) demonstrates a reduction from high to low, with nice even steps inbetween?
- Is it fair to use the same prior distribution for all tests? To my mind from reading the introduction, hypothesis 1 is “fairly sure”, #2 is “probable, but no actual evidence”, #3 is “fairly sure” and #4 (see above) is “near-certain”. Ought this difference in prior knowledge be reflected in the prior assumption for each test?
- In the analysis, what happened to the SNRs? There is no mention of “SNR” anywhere in the analysis section. Are they all being analyzed as individual points in the ANOVAs, or is a psychometric function to be fitted then that enters the ANOVA?
- Page 9 line 48 – does the Bayes criterion need be reached for all tests, or just one of them?
- Overall, why use a ANOVA in the first place? Both RTs and SNRs are on interval scales, so regression lines can be fitted to them, but ANOVAs don’t care about such ordering (if A-B-C vs D-E-F is significant, so is C-B-A vs F-E-D etc). Why wasn’t some form of regression analysis chosen?

Also, to be honest, 9 hypotheses is too many. It would be preferable to divide them into four primary hypotheses and the remainder as secondary. More importantly, hypothesis 4 – “hearing loss has an effect on speech intelligibility and the application of reverberation has an effect on speech intelligibility” – is surely entirely superfluous. I expect that there will not be any doubt in the field that the answers to both are Yes, so including them as hypotheses of the same status as the others dilutes the worth of the rest, and is certainly not of the level of scientific interest expected in a Royal Society journal. To my mind, it can be dealt with in a sentence in the experiment’s results (“as was expected, HI and reverb affected speech intelligibility”) before quickly moving onto results that do matter.

And for the record to answer the questions set

The scientific validity of the research question(s)

- Yes

-

The logic, rationale, and plausibility of the proposed hypotheses

- mostly yes, apart for hypothesis 4 and that nine in total is too many

-

The soundness and feasibility of the methodology and analysis pipeline (including statistical power analysis where applicable)

- mostly yes, apart from the comments above about the method & analysis.

-

Whether the clarity and degree of methodological detail would be sufficient to replicate exactly the proposed experimental procedures and analysis pipeline

- yes, apart from the issue of what happened to the SNRs in the analysis

-

Whether the authors provide a sufficiently clear and detailed description of the methods to prevent undisclosed flexibility in the experimental procedures or analysis pipeline

- mostly yes. To be honest, subsequent post-hoc analyses could always be done, but the primary ones as specified are clear enough to be followed.

-

Whether the authors have considered sufficient outcome-neutral conditions (e.g. positive controls) for ensuring that the results obtained are able to test the stated hypotheses

- I agree with the authors here, there’s no need for a positive control on this.

PS Its Plomp not Plomb, Butterworth not butterworth, and “and” not “et”

Reviewer: 3

Comments to the Author(s)

Review of registered report – stage 1 “Effects of reverberation on speech intelligibility in noise for hearing-impaired listeners”

The statistical analysis planned in this study appears extremely elaborated to me and I highly appreciate the effort that has already been invested by the authors and the editor. I learned a lot and will try to approximate to these standards at least a little in my research. I cannot really contribute to the statistical aspects of this study.

I think, however, that I can contribute to the audiological aspects of this study.

The research questions and the list of hypotheses are without any doubt very interesting and the method appears in principle adequate to answer these questions. My main concern, however, is that I do not think that the stimuli planned to be used in this study are adequate. The original FrMatrix test has been realized very elaborately so that it preserves quite natural prosody and coarticulation between words as far as this is possible for the Matrix sentences concept and the state of the art. Furthermore, the cautious selection of recordings and homogenization of the levels of the single words guarantees equal intelligibility across words, sentences and test lists. See for example the review of Kollmeier et al 2015 (DOI 10.3109/14992027.2015.1020971). The method proposed here, using rerecorded words, is similar to the first Matrix sentence test by Hagerman (1982, DOI 10.3109/01050398209076203) which was innovative for that time. Hagerman also recorded the words in isolation, but he did not adjust the levels for equal RMS. However, it is essential to understand that it is not an issue that the FrMatrix test has 280 sentences which is a smaller number as the total number required per listener in this study. It is virtually impossible to memorize these 280 sentences as they are all drawn from the same pool of 50 words and as they are always presented in random order. In fact none of the listeners will recognize that sentences are repeated and that most of the theoretically possible sentences never occur.

Nevertheless, there will be a learning issue in this study, but this learning effect is not due to the fact that the same sentences are used four or five times but due to the fact that each word will be used 112 to 140 times. This will cause a long term learning effect that cannot be avoided using the two 20 sentence lists for familiarization as recommended by Jansen et al. 2012 and as planned in this study. The initial familiarization reduces the learning effect by approximately 1 to 2 dB (see Hagerman 1982, Jansen et al 2012, and Kollmeier 2015). But the remaining learning effect of about one additional dB until 280 sentences is, for instance, reported by Hagerman and also by other authors. Unfortunately the learning effect continues even for longer measurements and across session as reported by Schlueter et al (2016, DOI 10.1177/2331216516669889). This long term-learning can be explained by the fact that the listeners develop to expert listeners for the words of the matrix and that they perform key word spotting (which is the best strategy) rather than real sentence recognition. The authors’ method to address this problem by rerecording the sentences does not decrease this learning effect, but it may increase it by facilitating the word spotting.

This long-term familiarization is compensated to a certain degree by randomization of the test conditions but nevertheless it will introduce a lot of noise into the data. I therefore recommend compensating the long-term learning effect on an average basis. An even better solution would be not to use Matrix sentences at all, as simple daily life sentences do not show a training effect due

to word spotting, but of course I see the problem that such test material with equal intelligibility across lists does not exist.

The rerecording of the words with neutral reflection is nonsensical and makes the problem that should be avoided even worse. I am sorry, that I have to express it so directly. The new construction of the recordings destroys prosody and coarticulation which are important cues for the building of acoustic streams and speech context which are main factors for dip listening. As dip listening is in the main focus of this study this has to be treated very cautiously.

Scaling the individual words to the same RMS might perhaps be convincing from a naïve technical point of view. But it is nonsensical from a speech processing point of view – sorry again. As different words are produced very differently they naturally differ in RMS as the averaging time is too short for a reliable and valid estimate of speaking effort. Instead it is recommended to record the words within sentences with cautious control of speaking effort by the talker and optionally adjust the levels of the sentences. Even better is the method applied in the matrix test concept like in the FrMatrix test, as it also adjusts the intelligibility between words improving the accuracy of each measurement. Therefore I strongly recommend to build on the experience of the quite high number of studies using the Matrix test concept and to use the original version of the FrMatrix test which is certainly better suited for this study than the material described in this manuscript.

The generation of the modulated noise is quite unnatural because such signals that are modulated like the broadband envelope of speech identically in all frequency bands hardly ever occur in real life. Speech is modulated slightly differently in different frequency bands. In order to increase ecological validity I recommend to use the modulated ICRA noises which show more speech-like modulations or to generate the modulated noise in a similar way as the ICRA noises (Dreschler et al, 2001).

The virtual room acoustics is very unnatural. Room with identical absorption for walls ceiling and floor do not exist. This contradicts the high effort for generating the rooms using CATT. This is not a severe problem for monaural measurements. But I could imagine that the authors want to extend their focus on binaural measurements. For such measurements the RIR would be really problematic.

It might be an issue that for each test condition 10 sentences containing the same pool of words are used. This depends on the manner of randomization mentioned below. This would have the consequence, that the later sentences in each block are biased by the words perceived in the previous trials as the best strategy is to repeat only words the listener hasn't heard so far in this list. In this case it is not a sentence intelligibility test but as a word spotting test combined with a short term memory span test. This has not sufficient ecological validity to investigate the effects of reverberation and dip listening on sentence recognition.

The selection of the five fixed SNRs for measurements has not been specified clearly enough. In order to guarantee that NH and HI can be tested at the same SNRs, the intelligibility function of the test has to be known. As it is new material the intelligibility function is unknown, so far. As the single words have not been adjusted for equal intelligibility but for equal RMS, the slope of the intelligibility function can be expected to be flatter than in the FrMatrix test, see Kollmeier 2015 where the relation between slope and intelligibility of single words is explained. In order to specify the SNRs for testing the intelligibility function or at least the expected range has to be specified. I see that pilot measurements will be done, but it should be made clear which target intelligibilities should be met and if it is possible to fulfill the requirements of overlapping intelligibility ranges between conditions and listener groups.

P8 L7-8: what does it mean that the test condition is tested on a trial-by-trial order? Is a trial here a 10 sentence-list or a single sentence. I hope that the latter is meant. Then the memory span issue mentioned above is not an issue. But please specify how the tests are divided into blocks.

I also want to mention that there is evidence supporting the hypothesis that matrix sentences are less affected by reverberation than daily life speech (see for example Rennies et al 2014, 10.1121/1.4897398). More data are required concerning this question, but I think that this point should be discussed in this study.

Minor comments:

P1 L52: white space in “the”

References are in French style, e.g. “Helfer et Wilber (1990)”.

P2 L39-58: due to the limitation of the scale and the non-linear intelligibility function it is problematic to compare intelligibility scores in %. If an intelligibility score decreases, for instance, from 100 to 80% it is a much stronger effect compared to a decrease from 60 to 40%. Later it is mentioned that RAUs will be used to compensate for this. It might be required to take this into account already here.

This also plays a role in P3 L12-13.

P41 L40: Why has the dip listening benefit to be evaluated at comparable SNRs? This is only one aspect. You can also argue that the dip listening benefit has to be sufficient high sensation level in both groups in order to assure that audibility is not the limiting factor in the dips, especially for the HI listeners.

P5 L53: Why do you plan to select the ear with the flatter hearing loss? One can also argue for selecting the better ear in order to reduce the effect of sensory deprivation due to the hearing loss of the weak ear.

In Beutelmann, Brand and Kollmeier (2010) different reverberation times and noises with different degrees of modulation have been tested with NH and HI listeners. The analysis is not as systematic as the analysis planned here, but there is at least one other study that investigated dip listening in reverberation in HI listeners.

P11: I think there is a typo in lower right field, second condition: “If $BF < 1/3$: the magnitude ...”

Author's Response to Decision Letter for (RSOS-210342.R0)

See Appendix B.

RSOS-210342.R1

Review form: Reviewer 1

Do you have any ethical concerns with this paper?

No

Recommendation?

Accept in principle

Comments to the Author(s)

The authors have addressed all my concerns adequately.

Decision letter (RSOS-210342.R1)

Dear Mr Cueille

On behalf of the Editor, I am pleased to inform you that your Manuscript RSOS-210342.R1 entitled "Effects of reverberation on speech intelligibility in noise for hearing-impaired listeners" has been accepted in principle for publication in Royal Society Open Science. The reviewers' and editors' comments are included at the end of this email.

You may now progress to Stage 2 and complete the study as approved. Before commencing data collection we ask that you:

- 1) Update the journal office as to the anticipated completion date of your study.
- 2) Register your approved protocol on the Open Science Framework (<https://osf.io/>) or other recognised repository, either publicly or privately under embargo until submission of the Stage 2 manuscript. Please note that a time-stamped, independent registration of the protocol is mandatory under journal policy, and manuscripts that do not conform to this requirement cannot be considered at Stage 2. The protocol should be registered unchanged from its current approved state, with the time-stamp preceding implementation of the approved study design.

Following completion of your study, we invite you to resubmit your paper for peer review as a Stage 2 Registered Report. Please note that your manuscript can still be rejected for publication at Stage 2 if the Editors consider any of the following conditions to be met:

- The results were unable to test the authors' proposed hypotheses by failing to meet the approved outcome-neutral criteria.
- The authors altered the Introduction, rationale, or hypotheses, as approved in the Stage 1 submission.
- The authors failed to adhere closely to the registered experimental procedures. Please note that any deviations from the approved experimental procedures must be communicated to the editor immediately for approval, and prior to the completion of data collection. Failure to do so can result in revocation of in-principle acceptance and rejection at Stage 2 (see complete guidelines for further information).
- Any post-hoc (unregistered) analyses were either unjustified, insufficiently caveated, or overly dominant in shaping the authors' conclusions.
- The authors' conclusions were not justified given the data obtained.

We encourage you to read the complete guidelines for authors concerning Stage 2 submissions at <https://royalsocietypublishing.org/rsos/registered-reports#ReviewerGuideRegRep>. Please especially note the requirements for data sharing, reporting the URL of the independently

registered protocol, and that withdrawing your manuscript will result in publication of a Withdrawn Registration.

Once again, thank you for submitting your manuscript to Royal Society Open Science and we look forward to receiving your Stage 2 submission. If you have any questions at all, please do not hesitate to get in touch. We look forward to hearing from you shortly with the anticipated submission date for your stage two manuscript.

on behalf of Professor Chris Chambers (Registered Reports Editor, Royal Society Open Science)
openscience@royalsociety.org

Associate Editor Comments to Author (Professor Chris Chambers):

The manuscript can now progress to Stage 1 IPA.

Reviewers' comments to Author:

Reviewer: 1

Comments to the Author(s)

The authors have addressed all my concerns adequately.

Author's Response to Decision Letter for (RSOS-210342.R1)

See Appendix C.

RSOS-210342.R2

Review form: Reviewer 1

Is the manuscript scientifically sound in its present form?

Yes

Are the interpretations and conclusions justified by the results?

Yes

Is the language acceptable?

Yes

Do you have any ethical concerns with this paper?

Yes

Have you any concerns about statistical analyses in this paper?

Yes

Recommendation?

Accept with minor revision

Comments to the Author(s)

My only substantial issue with the paper as it stands concerns the combination of RAUs with sigmoidal fitted functions. I would have expected the psychometric function to be linearised by the conversion to RAUs and so to have been fitted with a straight line.

p25, line 46 groups -> group

p26, lines 38-39. I think this sentence would be clearer, because "they" can be interpreted as meaning "noise interferers". Perhaps replace "they" with "stream segregation effects".

line 51 details -> detail.

p28, line 17. "and al." -> "et al."

line 46 Suggest replacing "more important" with "greater"

line 52. "aimed to test" - why not just "tested"?

p30, line 35 "informed-consent form"

p32, line 11, "to simulate" -> "simulation of"

line 32. "envelope" -> "envelopes"

p34, line 50. "assure" -> "ensure"

line 52. "insure" -> "ensure"

line 55. "a same" -> "the same"

line 57. "to not" -> "not to"

line 58. "allows to avoid" -> "avoids"

p34, line 6. "allows to test" -> "facilitates testing"

p35, line 18. I think this should be "Matrix tests are often used"

lines 28-34. If I understand this correctly, the lower asymptote of the psychometric function was allowed to vary between 0 and 0.05. This makes sense for the upper bound, for which there is an unknown lapse rate, but does not makes sense for the lower bound which should be fixed at chance performance calculated from the number of available stimulus permutations. But also, see below about whether there should be a sigmoidal fit in the first place for RAUs. It was not clear reading this section that the fitted parameter was RAUs rather than %-correct, since the input is referred to here simply as "results" (line 27).

p36, line 24. "were" -> "was"

line 37. "important"? I don't understand what this word is there for. It probably needs to be deleted.

p39, line 14. "infirm" -> "reject"?

p39, line 29 to p40, line 8. "allowed to test" -> "tested" (several instances)

lines 13 & 16. "allowed to determine" -> "determined"

line 31. "to obtain" -> "of obtaining"

line 44. "to conclude" -> "to reach a conclusion"

Figs. 4-6. I was a bit surprised to see sigmoidal fits to psychometric functions that have been converted to RAUs. I thought the idea of RAUs was that it linearised %-correct data.

line 28. delete "that is to say"

p49, line 20. "allowed to have a broader view" -> "permitted a broader view"

Decision letter (RSOS-210342.R2)

Dear Mr Cueille:

On behalf of the Editor, I am pleased to inform you that your Stage 2 Registered Report RSOS-210342.R2 entitled "Effects of reverberation on speech intelligibility in noise for hearing-impaired listeners" has been deemed suitable for publication in Royal Society Open Science subject to minor revision in accordance with the referee suggestions. Please find the referees' comments at the end of this email.

The reviewers and Subject Editor have recommended publication, but also suggest some minor revisions to your manuscript. We invite you to respond to the comments and revise your manuscript. Below the referees' and Editors' comments (where applicable) we provide additional requirements. Final acceptance of your manuscript is dependent on these requirements being met. We provide guidance below to help you prepare your revision.

Please submit your revised manuscript and required files (see below) no later than 7 days from today's (ie 22-Jul-2022) date. Note: the ScholarOne system will 'lock' if submission of the revision is attempted 7 or more days after the deadline. If you do not think you will be able to meet this deadline please contact the editorial office immediately.

on behalf of Professor Chris Chambers
 (Registered Reports Editor, Royal Society Open Science)
 openscience@royalsociety.org

Associate Editor Comments to Author (Professor Chris Chambers):

Associate Editor: 1

Comments to the Author:

One of the original Stage 1 reviewers was available to evaluate the Stage 2 manuscript, and I have decided that we can continue with an interim decision based on this review and my own reading of the manuscript.

As you will see, the evaluation is broadly positive, but there is one significant issue to resolve concerning the use of sigmoidal fits, which the reviewer argues are inappropriate. Normally the analytic procedures for a registered report cannot be altered at Stage 2 (because they are fixed at Stage 1), but unless I'm mistaken, this particular detail was only added to the manuscript after Stage 1 acceptance, therefore it must be resolved satisfactorily to achieve Stage 2 acceptance. It also needs to be made clear in a footnote associated with the methodological description (Section 2c, paragraph 3) *when* in the research workflow the decision to implement the particular model was made (either before or after data observation). This footnote must appear in the clean version of the manuscript as well as the tracked-changes version (and note that all substantial additions or alternations of the text other than tense changes or minor grammatical revisions must also be footnoted in both the clean version and tracked-changes version). If you believe that both your analytic approach and the one proposed by the reviewer are defensible then both should be reported in the manuscript as a robustness check.

Comments to Author:

Reviewer: 1

Comments to the Author(s)

My only substantial issue with the paper as it stands concerns the combination of RAUs with sigmoidal fitted functions. I would have expected the psychometric function to be linearised by the conversion to RAUs and so to have been fitted with a straight line.

p25, line 46 groups -> group

p26, lines 38-39. I think this sentence would be clearer, because "they" can be interpreted as meaning "noise interferers". Perhaps replace "they" with "stream segregation effects".

line 51 details -> detail.

p28, line 17. "and al." -> "et al."

line 46 Suggest replacing "more important" with "greater"

line 52. "aimed to test" - why not just "tested"?

p30, line 35 "informed-consent form"

p32, line 11, "to simulate" -> "simulation of"

line 32. "envelope" -> "envelopes"

p34, line 50. "assure" -> "ensure"

line 52. "insure" -> "ensure"

line 55. "a same" -> "the same"

line 57. "to not" -> "not to"

line 58. "allows to avoid" -> "avoids"

p34, line 6. "allows to test" -> "facilitates testing"

p35, line 18. I think this should be "Matrix tests are often used"

lines 28-34. If I understand this correctly, the lower asymptote of the psychometric function was allowed to vary between 0 and 0.05. This makes sense for the upper bound, for which there is an unknown lapse rate, but does not makes sense for the lower bound which should be fixed at chance performace calculated from the number of available stimulus permutations. But also, see below about whether there should be a sigmoidal fit in the first place for RAUs. It was not clear reading this section that the fitted parameter was RAUs rather than %-correct, since the input is referred to here simply as "results" (line 27).

p36, line 24. "were" -> "was"

line 37. "important"? I don't understand what this word is there for. It probably needs to be deleted.

p39, line 14. "infirm" -> "reject"?

p39, line 29 to p40, line 8. "allowed to test" -> "tested" (several instances)

lines 13 & 16. "allowed to determine" -> "determined"

line 31. "to obtain" -> "of obtaining"

line 44. "to conclude" -> "to reach a conclusion"

Figs. 4-6. I was a bit surprised to see sigmoidal fits to psychometric functions that have been converted to RAUs. I thought the idea of RAUs was that it linearised %-correct data.

line 28. delete "that is to say"

p49, line 20. "allowed to have a broader view" -> "permitted a broader view"

===PREPARING YOUR MANUSCRIPT===

one version should clearly identify all the changes that have been made (for instance, in coloured highlight, in bold text, or tracked changes);

===PREPARING YOUR REVISION IN SCHOLARONE===

-- If you are requesting an article processing charge waiver, you must select the relevant waiver option (if requesting a discretionary waiver, the form should have been uploaded, see 'File upload' above).

-- If you have uploaded any electronic supplementary (ESM) files, please ensure you follow the guidance at <https://royalsociety.org/journals/authors/author-guidelines/#supplementary-material> to include a suitable title and informative caption. An example of appropriate titling and captioning may be found at https://figshare.com/articles/Table_S2_from_Is_there_a_trade-off_between_peak_performance_and_performance_breadth_across_temperatures_for_aerobic_scope_in_teleost_fishes_/3843624.

Author's Response to Decision Letter for (RSOS-210342.R2)

See Appendix D.

Decision letter (RSOS-210342.R3)

Dear Mr Cueille:

I am pleased to inform you that your manuscript entitled "Effects of reverberation on speech intelligibility in noise for hearing-impaired listeners" is now accepted for publication in Royal Society Open Science.

Please remember to make any data sets or code libraries 'live' prior to publication, and update any links as needed when you receive a proof to check - for instance, from a private 'for review' URL to a publicly accessible 'for publication' URL. It is also good practice to add data sets, code and other digital materials to your reference list.

Royal Society Open Science is a fully open access journal. A payment may be due before your article is published. Our partner Copyright Clearance Center's RightsLink for Scientific Communications will contact the corresponding author about your open access options from the email domain @copyright.com (if you have any queries regarding fees, please see <https://royalsocietypublishing.org/rsos/charges> or contact authorfees@royalsociety.org).

on behalf of Professor Professor Chris Chambers (Subject Editor).

Follow Royal Society Publishing on Twitter: @RSocPublishing
Follow Royal Society Publishing on Facebook:
<https://www.facebook.com/RoyalSocietyPublishing/>
Read Royal Society Publishing's blog:
<https://royalsociety.org/blog/blogsearchpage/?category=Publishing>

Appendix A

Dear editor,

You can find our answers to your comments below.

1. Please ensure that there is a direct correspondence between the proposed hypotheses, the sampling plans, and the proposed statistical tests in every case. At present these links are somewhat present but are limited to prose descriptions and too vague. To make the review process as straightforward as possible, please include a design table for the proposed study in the Method section of the main text based on Section 9 of this template: <https://osf.io/93znh/> You can find published examples of how these appear in these submissions (from different fields, but the principles are the same): <https://osf.io/xhdpu>, <https://osf.io/ym8gc>, <https://osf.io/g9dxb/>. Please ensure that each prediction is associated with a statistical sampling plan, a specific test (or set of tests) on specifically defined variables, and a comprehensive interpretative plan given different outcomes (i.e. a precommitment to different conclusions given different results).

The design table was included at the end of the article. A graph was also made to better illustrate the interpretative plan and the different conclusions and was put in the supplementary materials. Your remarks also motivated us to detail more our hypotheses and our statistical tests. This led us to add several hypotheses and tests to be more precise in our analysis. All the hypotheses and tests are presented in the design table, the supplementary graph and the cover letter.

2. Related to point 1, please consider whether the prespecified analyses themselves are sufficiently precise to confirm or disconfirm the hypotheses. For some predictions the authors predict interactions, but interactions can be supported multiple different datapatterns, not all of which may be consistent with the respective prediction. For example: "...whether HI and NH listeners are affected differently by reverberation when it is applied on both sources." Is any differential effect sufficient confirmatory evidence? Is the key confirmatory support for the predictions, in the end, supported by specific pairwise differences between conditions, or combinations thereof? The authors mention post hoc tests, but it is not clear whether these are necessary to confirm/disconfirm the hypotheses or are more exploratory.

For the first question (are HI listeners more affected by reverberation when it is applied only to speech?), we do three hypotheses:

- 1: the temporal smearing of the speech caused by reverberation impairs more HI listeners than NH listeners.
- 1-1: the temporal smearing of the target speech impairs intelligibility at lower levels of reverberation for HI listeners compared to NH listeners (that is to say HI listeners are more affected than NH listeners at low RTs).
- 1-2: Hypothesis 1-2: at high levels of reverberation, HI listeners are more impaired than NH listeners by the temporal smearing of the speech (that is to say, HI listeners are more affected than NH listeners at high RTs).

The Bayesian ANOVA allows to answer hypothesis 1: if there is a group x reverberation time (RT) interaction and the scores of the HI listeners are more reduced under reverberation, it can only mean that reverberation impairs more the HI listeners. In this case we claim full support for hypothesis 1.

If hypothesis 1 is confirmed we proceed with the post hoc tests. To answer 1-1, we do pairwise comparisons between RT = 0 s and the other RTs. This allows to determine the lowest RT at which each group is first affected. If the HI listeners have a score difference at a lower RT than the NH group, we claim full support for hypothesis 1-1. To answer 1-2, we do a t-test to compare the difference in

intelligibility scores between RT = 0 s and RT = 1.5 s between the two groups. If the Bayes Factor is superior to 3 for this comparison we claim full support for 1-2.

For the second question (are HI listeners affected differently by reverberation when it is applied only to modulated noise?), we do only one hypothesis: reverberation affects the dip listening benefit more for NH listeners than for HI listeners (hypothesis 2). To answer it, a Bayesian ANOVA is necessary. It allows to determine whether the interaction between reverberation and noise modulation is different for both groups. If the ANOVA detects a RT x group x noise interaction, we claim full support for hypothesis 2. Since it is harder to do precise hypotheses for the interaction between dip listening, reverberation and hearing loss, the post hoc tests from this part have been removed.

For the third question (are HI and NH listeners affected differently by reverberation when it is applied to noise and speech?), we do three hypotheses:

- 3: the overall (monaural) effect of reverberation impairs more HI listeners than NH listeners.
- 3-1: overall, reverberation impairs intelligibility at lower levels of reverberation for HI listeners compared to NH listeners.
- 3-2: overall, high reverberation levels impair intelligibility more for HI listeners compared to NH listeners.

The Bayesian ANOVA allows to answer hypothesis 3: if there is a group x reverberation time (RT) interaction and the scores of the HI listeners are more reduced under reverberation, it can only mean that the two groups are affected differently by this application of reverberation. In this case we claim full support for hypothesis 3.

If hypothesis 3 is confirmed we proceed with the post hoc tests. To answer 3-1, we do pairwise comparisons between RT = 0 s and the other RTs. This allows to determine the lowest RT at which each group is first affected. If the HI listeners have a score difference at a lower RT than the NH group, we claim full support for hypothesis 3-1. To answer 3-2, we do a t-test to compare the difference in intelligibility scores between RT = 0 s and RT = 1.5 s between the two groups. If the Bayes Factor is superior to 3 for this comparison we claim full support for 3-2.

For the fourth question (how do temporal smearing of the speech and dip listening decrease contribute to overall effect of reverberation?), we do three hypotheses:

- 4: reverberation has an effect on speech intelligibility.
- 4-1: dip listening decrease caused by reverberation contributes to the overall effect of reverberation for NH listeners, HI listeners or both NH and HI listeners.
- 4-2: temporal smearing of the speech caused by reverberation contributes to the overall effect of reverberation for NH listeners, HI listeners or both NH and HI listeners.

The Bayesian ANOVA considering all the common conditions allows to answer 4. If it detects a main effect of the application type we claim full support for hypothesis 4 and proceed with the post hoc tests.

For hypotheses 4-1 and 4-2 we do two supplementary ANOVAs.

3. I find the sampling plan to be a slightly strange mix of Bayesian and frequentist methods. Are the authors employing a Bayes Factor Design Analysis (BFDA; see <https://link.springer.com/article/10.3758/s13423-017-1230-y>)? This may be a more coherent approach. I suspect the current method will draw criticism from statistical reviewers. Please also make

clear what prior will be used in the Bayesian analyses and why. Minor point: "bayesian" should be capitalised throughout, and it is "Bayes factor", not "bayesian factor".

As you suggested the sampling plan has been changed for a BFDA, more specifically a SBF+maxN method, based on the work of Schönbrodt et Wagenmakers (2018).

Because we lacked sufficient data to establish an informative prior, we decided to use a Cauchy prior distribution as a default prior, with a scale value of 0.5.

Since no study tested HI listeners with reverberation and modulated noise it was not possible to make a sampling plan for the analyses regarding the hypotheses 2 and 4 and only the hypotheses 1 and 3 were studied.

With this method it has been estimated that the study had a probability of 0.02 to stop because reaching the maximum number of listeners, a probability of 0.80 to detect an effect if it was present, a probability of 0.86 to detect an absence of effect if it was absent, and a probability of 0.155 to obtain misleading evidence. The mean number of listeners is estimated at 36.

4. One of the key criteria that reviewers are asked to assess in Stage 1 RRs is "Whether the authors have considered sufficient outcome-neutral conditions (e.g. absence of floor or ceiling effects; positive controls; other quality checks or sanity checks) for ensuring that the results obtained are able to test the stated hypotheses". Such tests should be orthogonal of the main hypotheses. Your study does not obviously propose such tests (e.g. confirming, independent of the main outcomes, that the data quality was sufficient to be diagnostic about the hypotheses). Please consider whether such positive controls or sanity checks are necessary, and if possible, how they might be included in the design.

In order to avoid floor or ceiling effects, we had already planned to do a preliminary experiment to determine the SNRs to test. We had also planned to measure several SNRs, in order have the psychometric function of all listeners and determine the most appropriate SNR to compare HI and NH performance.

In addition to these elements, we have decided that the HI listeners will have a linear frequency-dependent gain applied to the stimuli before presentation. This gain will be set on an individual basis according to the NAL-RP prescription rule. This linear amplification allows to partly compensate for the audibility loss and should insure that the score difference at a same SNR is not too large between the HI and the NH groups.

We decided to not include positive controls in the study. Indeed, it would be necessary to test a large number of listeners to get a sufficient statistical power for the positive controls. Furthermore comparable conditions have already been tested in previous studies (Duquesnoy et Plomp, 1980; Harris et Swenson, 1990; Jensen et Bernstein, 2019). This allows us to be sure that the effects we want to study will be present with the conditions we intend to test.

5. Authors usually prepare their Stage 1 RR submissions in future tense to avoid reviewers from concluding (mistakenly) that the work has already been done with the results merely redacted. If the authors prefer past tense (which does have the advantage of efficiency at Stage 2), then I stringly suggest adding a cover note to the top of the manuscript making it clear to reviewers that although the rationale and methods are for expediency written in past tense, none of the research has yet been undertaken.

In regard to this comment we put the method part of the submission in the future tense. The only elements remaining in the past tense are elements that have already been done, such as the recording of the words, the RIRs generation and the sampling plan.

Conclusion

To conclude, as a first year PHD student I am highly motivated to submit this manuscript as a registered report, to help promote good practice in science during my PHD project.

Best regards,

Raphaël Cueille

Appendix B

Response to the reviewers.

Dear reviewers,

We would like to thank you for the time you took to comment our draft. Your remarks were really constructive and we tried to take them into account as much as possible. This really helped us to improve this experiment. Some of your comments highlighted important mistakes in our protocol and this has really showed us the benefits of registered reports.

You can find below a summary of the main changes that your remarks led us to do.

-The comments from the reviewer 3 led us to change our speech and noise material. Instead of doing our own recording of the FrMatrix we decided to use the original recordings, in order to avoid the destruction of prosody and coarticulation, limit the risk of word spotting, as well as benefit from the equalization of intelligibility across word, sentences and test lists.

To make sure that the modulated noise had speech-like modulations, we decided to create two-voice vocoded speech. We used a 3-channel vocoder, with an envelope low-pass filter-cut off frequency of 60 Hz, as also done by Rana and Buchholz (2018). To create the vocoded speech, we used the sentences from Raake and Katz (2006), which have more unpredictable modulations than the sentences of the FrMatrix. To make sure that the stationary noise and the vocoded speech have similar long-term spectra, the stationary noise was created with the sentences from Raake and Katz (2006). The stationary and modulated noises were also filtered so that they have the same long-term spectrum as the target sentence.

-One minor comment from the reviewer 3 made us notice that we did not put in RAUs some results of the literature. When it is done, it appears that the overall monaural effect of reverberation is less different for NH and HI listeners compared to what was previously thought. This comment also led us to notice a mistake in our power analysis: instead of converting the scores in RAUs, we directly did the power analysis with the scores in percentages. This led to a ceiling effect and an overestimation of the group x RT interaction. When the RAU conversion is done, the difference between the two groups is so small that the Bayesian ANOVA confirms an absence of interaction RT x group for 80% of the simulations.

These elements led us to rethink our hypothesis 3-1, which is now as follows: the overall (monaural) effect of reverberation impairs HI listeners as much as NH listeners. This change does not affect the rest of the hypotheses. Even if the global effect of reverberation is similar for HI and NH listeners, both groups might be affected differently either by the temporal smearing of the speech or the dip listening decrease.

-The comments from the reviewers 2 and 1 made us rethink our hypotheses and analyses. The hypothesis 4 and 4-2 appear to be near certain and are not as interesting as the other hypotheses. We decided to remove them from the analysis plan. The tests linked to these hypotheses will still be done, but only as a form of positive control to confirm these near certain hypotheses. However, the hypothesis 4-1 seems interesting to us: it might be useful to determine whether the dip listening decrease contributes to the overall effect of reverberation, since it is a priori a smaller effect than the temporal smearing of the speech. Instead of doing ANOVAs across applications of reverberation, we decided to do two supplementary ANOVAs on the "speech+noise" condition, to evaluate this effect for HI and NH listeners.

Removing the hypothesis 4 also made us rethink the hypotheses 3-1 and 3-2. They seemed redundant with 1-1 and 1-2, which tested similar effects. This motivated us to remove them.

To summarize, our new set of hypotheses and tests is as follows:

1: same as before

1-1: same as before.

1-2: same as before.

2: same as before.

3-1: the overall (monaural) effect of reverberation impairs HI listeners as much as NH listeners. To evaluate it we do an ANOVA with the results of both groups in the "speech+noise" condition, as we previously did for hypothesis 3. However, instead of expecting the presence of a RT x group interaction, we make the hypothesis that there will not be an interaction.

3-2-a: In the case of NH listeners the dip listening decrease has a significant effect on intelligibility when reverberation is applied to speech and noise. To evaluate it we do an ANOVA with the results of the NH group in the "speech + noise" condition and look at the RT x noise interaction.

3-2-b: In the case of HI listeners the dip listening decrease has a significant effect on intelligibility when reverberation is applied to speech and noise. To evaluate it we do an ANOVA with the results of the HI group in the "speech + noise" condition and look at the RT x noise interaction.

This leaves us with 7 hypotheses instead of 10. Among these, there are three main hypotheses (1, 2, 3-1) and four which are more secondary (1-1, 1-2, 3-2-a and 3-2-b).

These modifications of the hypotheses and analyses led us to remove the RT = 0.5 s in the "speech + noise" and "noise only" conditions. Since we are now only interested in showing the presence of an effect, only the extreme level of reverberation is of interest.

- A comment from the reviewer 2 comment motivated us to do a modulation analysis for the noise and speech signals. To do this we generated the modulation spectra of the speech and noise signals by computing the envelope of the signal, dividing this envelope by its mean and calculating its power spectrum in octave bands between 1 and 32 Hz. We found that the anechoic speech had similar modulation levels to the speech with RT = 0.15 s (that is to say, the speech convoluted with the RIR of the room with the absorption coefficients of mineral wool). We decided to test the RT = 0.15 s condition instead of the anechoic condition. This allows to have all the signals convoluted with BRIRs varying only due to a single parameter (the gain applied to the absorption coefficients of the room). We also found that the RT = 0.2 s condition was not very useful, as the modulation spectrum obtained was very close to the ones obtained with RT = 0.15 s and with the anechoic condition. We decided to remove this condition and add another intermediary RT (1.1 s). To resume, instead of using 0 s, 0.2 s, 0.5 s, 0.8 s, and 1.5 s we now test 0.15 s, 0.5 s, 0.8 s, 1.1 s and 1.5 s. All the modulation spectra can be found in the new draft of the registered report (Figure 1 and Figure 2). You can find below another version of the speech modulation spectra, with the discarded condition (RT = 0.2 s).

You can find below more detailed answers in response to each of your comments. Your comments appear in italics and our answers are in roman.

Comments to Author:

Reviewer: 1

Comments to the Author(s)

General comments.

The proposed experiment examines the interacting effects of masker modulation and room reverberation on speech intelligibility for hearing-impaired (HI) listeners (compared to normally hearing (NH))

listeners). The design and methods of analysis look well-suited to answering the set of proposed hypotheses. The authors have negotiated some difficult design problems that lie unaddressed in most previous work in this area. These problems are described in the Introduction to some extent, but perhaps not as directly as they might be. One problem with many of the studies described is that those studies report percentage changes in word identification, but the relative sizes of such changes are often dependent upon the baseline level of intelligibility (where modulation and/or reverberation are absent). Since the baseline is different for NH and HI listeners, intelligibility is nearer floor or nearer ceiling in comparison conditions, making the changes in percentage difficult to interpret. Many studies avoid floor and ceiling effects by measuring shifts in speech reception threshold, but these studies consequently compare NH and HI performance at different signal-to-noise ratios. One study that they describe (Jepsen and Bernstein) elegantly skirted these problems by first equalising NH and HI intelligibility through variations in the size of the response set. The present study aims to achieve the same end by transforming percent correct scores at the same signal-to-noise ratios into rationalised arcsine units which reduce the influence of floor and ceiling effects. I have to say that I do prefer the Jensen and Bernstein approach, and I wonder whether it might be possible to adopt it in the present study. It might be straightforward to change the number of different tokens used in the FrMatrix test so that NH and HI performance is equalised, but it would require some pretesting to calibrate the required numbers.

The method used by Jensen and Bernstein to test NH and HI listeners at similar SNRs seems very interesting to us as well. However, the method that we propose here should allow us to achieve similar results. With the use of the NAL-RP amplification, the rationalized arcsine units, as well as the pre-tests, we should be able to test HI and NH listeners in the same SNR range without being affected by floor and ceiling effects. Using the Jensen and Bernstein method, in other words changing the number of possible answers for HI and NH listeners, would also modify the FrMatrix test. This is something we would like to avoid, especially regarding the remarks from the reviewer 3.

Beyond this, the study will examine for the first time the relative influences of reverberation on the target and interfering noise in HI listeners, as well as the interaction with masker modulation. This all looks like an excellent plan.

Minor points (page #s from top left corner).

P2, L25. This may need rephrasing. There are also monaural effects on the processing of F0 differences.

It is true that we omitted the processing of F0 differences and only reported the temporal smearing of the speech and noise as monaural effects. Since this study focuses on speech in noise conditions, we decided to only rephrase "two main monaural effects have been reported" into "two main monaural effects are studied here".

P4, L10 and elsewhere. It is commonplace to specify reverberation by the reverberation time, but it is, from the point of view of the effect on temporal smearing, a very tangential parameter. The important factor is the direct-to-reverberant ratio. For a fixed room and source/receiver geometry (as used in the proposed expt.) direct-to-reverberant ratio and reverberation time will vary in the same way, so there is internal consistency. However, comparisons across experiments, which may use different room sizes and/or source distances, become invalid unless transformed into direct-to-reverberant ratio using details of the geometry. It would be good to at least adopt the habit of reporting both RT and D/R in order to facilitate such comparisons. On this particular line, RT=0.25 s and RT=0.5 s are described as "low", but I cannot tell from these numbers alone whether the D/R was low.

Thanks for bringing this to our attention. Unfortunately, the studies that we present in the introduction only mention the RTs they used and not their D/R ratios, so it is unknown whether the differences in room

geometries had an effect here. The room sizes used by the studies mentioned in the Introduction have been added and Table 2 now includes the D/R ratios in our experimental conditions.

P4, L39-41. The Jensen and Bernstein method could be described in more detail here.

More details were added in the introduction regarding the method they used to test both groups at similar SNRs and audibility.

P5. I found it hard to digest the hypotheses before the following paragraph summarizing the method. I was a bit puzzled by hypotheses 4,4-1,4-2, since 1, 2 and 3 looked as though they were contingent on the same things. Maybe these are fallback hypotheses.

We agree with you and the reviewer 2 that there were too many hypotheses. As explained above, we decided to remove 4 and 4-2, which are not as interesting as the other hypotheses. We kept 4-1, but a different test is used to verify it. More details can be found about this in our answers to the reviewer 2 and in the new draft of the registered report. The number of hypotheses falls down from 10 to 7 in this new version of the manuscript.

P7. Again, it would be nice to see the D/Rs for these conditions.

The D/R ratios of our simulated rooms have been added in the registered report (Table 2).

P7, last para. I was quite confused at this point, because the use of rationalized arc-sine units had not yet been mentioned, and I didn't see how these things could be legitimately taken from the data.

It is true that the RAUs were only mentioned in the data analysis part. In the new draft of the registered report the RAUs are presented in the introduction.

Reviewer: 2

Comments to the Author(s)

This is a carefully designed study that ought to add knowledge to the question of the effect of reverberation on hearing-impairment. It is encouraging for the future of our field that a PhD student wishes to write a registered report – well done.

However certain choices need further exposition. Now, none of them are technically incorrect (excepting possibly the number-of-trials and the assumed priors), but the questions are raised to encourage reflection on each, as there may be better choices for each one that will improve the rigor and potential importance of the research:

- *Why a room of 5m x 9m x 3 m, and why these particular positions within it (which reduces to the question of will this room only give results of particular interest to rooms just like this, or does it give generalizable results of much more interest?)*

The configuration of the room and the positions were only chosen so that both the speaker and the listener were not in corners of the room. Our simulation is somehow artificial, but it allows for a crucial experimental point that is to control reverberation in a single and simple way, varying a unique (frequency independent) gain applied to the absorption coefficients. This implies that the speech and noise reflections have the same temporal pattern across conditions, and only the energy of these reflections change. Generalizing the results should be the goal of a follow-up study, which could for example test several real RIRs.

- *Why only 10 trials per SNR? This really worries me, I'd have used 20 at least if not 30. Ten really is very low, and I am not convinced that it will give data of sufficient reliability to be able to properly answer the questions at hand.*

10 trials per SNR can indeed seem low. However, it is important to note, that most studies using the FrMatrix do 20 trials to measure one SRT, in other words 20 trials per condition. In this experiment for each condition we test 5 SNRs per condition and use these 5 SNRs to do a fitting of the psychometric curve of the listener. In other words, we do 50 trials to get the psychometric curve of the listener (corresponding to a single SRT).

- *Why a presentation level of just 60 dB? That's fairly quiet I would have thought.*

In the past weeks we tried to test ourselves and found that it will be necessary to test at relatively low SNRs (down to -20 dB) in order to get the entire psychometric functions. This implies that the noise will be up to 20 dB higher than the presentation level. Furthermore, for the HI listeners the NAL-RP amplification can raise the sound level by up to 20 dB. It is thus necessary to have a low presentation level in order to avoid loudness or clipping problems. In fact, we found that a presentation level of 50 dB is necessary to make sure that the stimuli are not uncomfortable for the HI listeners.

This presentation level is indeed very low. To make sure that the listeners can hear the sentences, we decided to add an audibility test at the beginning of the experiment, which consist of a list of twenty sentences in quiet. If the corresponding audibility score of the listener is below 90 %, we will test him/her at a higher presentation level.

We also added a loudness test to make sure that the stimuli were not uncomfortable for the listener.

More details about these two pre-tests can be found in the new draft of the registered report.

- *Why don't do $rt = 0.2$ s and 0.8 s in the Noise & Speech-Noise conditions (table 3). I would have included them to complete the cells in the experiment.*

In the case of the "noise only" condition, we are only making a hypothesis regarding the presence of an effect. The intermediary conditions are not very useful here, and only the lowest and highest RTs are of interest. Thus, we decided to not add these two RTs for the noise condition. The $RT=0.5$ s condition was originally kept in order to have more common points for the ANOVAs concerning all the conditions (hypotheses 4, 4-1 and 4-2). However, since one of your comments motivated us to remove these ANOVAs, the $RT = 0.5$ s in the "noise only" condition is now useless and we decided to remove it.

The comment that you made below regarding hypothesis 4 also led us to rethink the "speech + noise" condition. With our new hypotheses, the intermediary RT condition become unnecessary for this condition and we removed it. More details about this can be found below, in the new draft, in the design table and in the illustrations of the possible outcomes.

- *Why chose these particular RTs? Has a quantitative analysis of the modulations been done, that (ideally) demonstrates a reduction from high to low, with nice even steps in-between?*

We did not do an analysis of the modulations, thanks for bringing this to our attention. The intermediary RTs were chosen because Harris and Swenson (1990) did the hypothesis that HI listeners had a more important intelligibility decrease around $RT = 0.5$ s. This motivated us to test RT values around 0.5 s (0.3 s above and below this value) to have a more precise understanding of the effect of reverberation on HI listeners around these reverberation times.

Your comment motivated us to do a modulation analysis for the speech and noise signals. This led us to remove one of the lowest RT and add a higher one. You can find more details about this above (in our summary of the main changes), and in the revised draft.

• *Is it fair to use the same prior distribution for all tests? To my mind from reading the introduction, hypothesis 1 is "fairly sure", #2 is "probable, but no actual evidence", #3 is "fairly sure" and #4 (see above) is "near-certain". Ought this difference in prior knowledge be reflected in the prior assumption for each test?*

The previous introduction was a little misleading regarding the hypotheses. Hypotheses 1 and 3 appear fairly sure in the introduction because the scores presented appeared in percent. When converted in RAUs, the hypotheses 1 and 3 and their post hoc appear less certain. As explained below, we decided to remove hypothesis 4. Thus, we would say that:

- hypothesis 1 is probable, but the evidence might be misleading.
- hypothesis 1-1 is probable, but the evidence might be misleading.
- hypothesis 1-2 is probable, but the evidence might be misleading.
- hypothesis 2 is probable, but there is no evidence.
- hypothesis 3-1 is probable, but the evidence might be misleading.
- hypothesis 3-2-a is probable, but there is no evidence.
- hypothesis 3-2-b is probable, but there is no evidence.

We do not think this is sufficient to establish different priors for each test.

• *In the analysis, what happened to the SNRs? There is no mention of "SNR" anywhere in the analysis section. Are they all being analyzed as individual points in the ANOVAs, or is a psychometric function to be fitted then that enters the ANOVA?*

With the 5 SNRs we intend to fit a psychometric curve to each listener and condition, with the psignifit toolbox (version 4). For the analysis our goal is to take a common SNR for all the psychometric curves and do all the tests at this one SNR. This was not explained clearly in the previous draft, thanks for bringing this to our attention. We tried to clarify this point in the new draft of the registered report.

• *Page 9 line 48 – does the Bayes criterion need be reached for all tests, or just one of them?*

We intend to keep testing listeners until a decisive Bayes Factor has been reached for all the main hypotheses (1, 2, 3-1, 3-2-a and 3-2-b) or until the maximum number of listeners is reached.

• *Overall, why use an ANOVA in the first place? Both RTs and SNRs are on interval scales, so regression lines can be fitted to them, but ANOVAs don't care about such ordering (if A-B-C vs D-E-F is significant, so is C-B-A vs F-E-D etc). Why wasn't some form of regression analysis chosen?*

In our new analysis plan, the RTs are on an interval scale only for the "Speech only" condition. For this condition we have no clear hypothesis regarding the relation between RT and intelligibility decrease. Because of this we decided to stick to the ANOVA as a main test and t-tests as post hoc tests. The ANOVA allow to detect whether there is an effect and the t-tests allow to determine its magnitude and at which RT value the effect happens. We thought it allowed us to reach similar results compared to a regression analysis.

Also, to be honest, 9 hypotheses is too many. It would be preferable to divide them into four primary hypotheses and the remainder as secondary. More importantly, hypothesis 4 – "hearing loss has an effect on speech intelligibility and the application of reverberation has an effect on speech intelligibility" – is surely entirely superfluous. I expect that there will not be any doubt in the field that the answers to both are Yes, so including them as hypotheses of the same status as the others dilutes the worth of the rest, and is certainly not of the level of scientific interest expected in a Royal Society journal. To my mind, it can be dealt with in a sentence in the experiment's results ("as was expected, HI and reverb affected speech intelligibility") before quickly moving onto results that do matter.

We agree with you, thanks for bringing this to our attention. We have decided to rethink our analysis plan. Two of the near certain hypotheses have been removed and some statistical tests and conditions tested have been adapted. You can find more details about this in the summary of the main changes of the experiment, at the beginning of this document.

PS Its Plomp not Plomb, Butterworth not butterworth, and "and" not "et"

This has been modified.

Reviewer: 3

The research questions and the list of hypotheses are without any doubt very interesting and the method appears in principle adequate to answer these questions. My main concern, however, is that I do not think that the stimuli planned to be used in this study are adequate. The original FrMatrix test has been realized very elaborately so that it preserves quite natural prosody and coarticulation between words as far as this is possible for the Matrix sentences concept and the state of the art. Furthermore, the cautious selection of recordings and homogenization of the levels of the single words guarantees equal intelligibility across words, sentences and test lists. See for example the review of Kollmeier et al 2015 (DOI 10.3109/14992027.2015.1020971). The method proposed here, using rerecorded words, is similar to the first Matrix sentence test by Hagerman (1982, DOI 10.3109/01050398209076203) which was innovative for that time. Hagerman also recorded the words in isolation, but he did not adjust the levels for equal RMS.

However, it is essential to understand that it is not an issue that the FrMatrix test has 280 sentences which is a smaller number as the total number required per listener in this study. It is virtually impossible to memorize these 280 sentences as they are all drawn from the same pool of 50 words and as they are always presented in random order. In fact, none of the listeners will recognize that sentences are repeated and that most of the theoretically possible sentences never occur.

Nevertheless, there will be a learning issue in this study, but this learning effect is not due to the fact that the same sentences are used four or five times but due to the fact that each word will be used 112 to 140 times. This will cause a long-term learning effect that cannot be avoided using the two 20 sentence lists for familiarization as recommended by Jansen et al. 2012 and as planned in this study. The initial familiarization reduces the learning effect by approximately 1 to 2 dB (see Hagerman 1982, Jansen et al 2012, and Kollmeier 2015). But the remaining learning effect of about one additional dB until 280 sentences is, for instance, reported by Hagerman and also by other authors. Unfortunately, the learning effect continues even for longer measurements and across session as reported by Schlueter et al (2016, DOI 10.1177/2331216516669889). This long term-learning can be explained by the fact that the listeners develop to expert listeners for the words of the matrix and that they perform key word spotting

(which is the best strategy) rather than real sentence recognition. The authors' method to address this problem by rerecording the sentences does not decrease this learning effect, but it may increase it by facilitating the word spotting.

Thank you very much for taking the time to explain all these elements to us. We have decided to stick to the original sentences of the FrMatrix and to not record other sentences ourselves.

This long-term familiarization is compensated to a certain degree by randomization of the test conditions but nevertheless it will introduce a lot of noise into the data. I therefore recommend compensating the long-term learning effect on an average basis. An even better solution would be not to use Matrix sentences at all, as simple daily life sentences do not show a training effect due to word spotting, but of course I see the problem that such test material with equal intelligibility across lists does not exist.

Thank you for bringing this to our attention. However, it seems complicated to us to compensate this learning effect in this case: to our knowledge there is no study that determined this long-term training effect for the FrMatrix in terms of intelligibility percentages and for the SNRs that we intend to test. We think that randomization of the test conditions along with a sufficient number of listeners participating to the study should be sufficient to compensate for the noise induced in the data by this potential learning effect. Finally, we agree with the reviewer that lists of French daily life sentences with equal across list does not exist.

The rerecording of the words with neutral reflection is nonsensical and makes the problem that should be avoided even worse. I am sorry, that I have to express it so directly. The new construction of the recordings destroys prosody and coarticulation which are important cues for the building of acoustic streams and speech context which are main factors for dip listening. As dip listening is in the main focus of this study this has to be treated very cautiously.

Scaling the individual words to the same RMS might perhaps be convincing from a naïve technical point of view. But it is nonsensical from a speech processing point of view – sorry again. As different words are produced very differently they naturally differ in RMS as the averaging time is too short for a reliable and valid estimate of speaking effort. Instead it is recommended to record the words within sentences with cautious control of speaking effort by the talker and optionally adjust the levels of the sentences. Even better is the method applied in the matrix test concept like in the FrMatrix test, as it also adjusts the intelligibility between words improving the accuracy of each measurement. Therefore, I strongly recommend to build on the experience of the quite high number of studies using the Matrix test concept and to use the original version of the FrMatrix test which is certainly better suited for this study than the material described in this manuscript.

Again, thank you very much for taking the time to give us all these elements regarding the matrix tests.

The generation of the modulated noise is quite unnatural because such signals that are modulated like the broadband envelope of speech identically in all frequency bands hardly ever occur in real life. Speech is modulated slightly differently in different frequency bands. In order to increase ecological validity, I recommend to use the modulated ICRA noises which show more speech-like modulations or to generate the modulated noise in a similar way as the ICRA noises (Dreschler et al, 2001).

Thank you for bringing this to our attention. In order to have more speech-like modulations, we decided to use 2-voice vocoded speech created with a low number of channels (3-channel vocoder, using an envelope low-pass filter-cut off frequency of 60 Hz), as also done by Rana and Buchholz (2018). You can find more details about this above (in the summary of the main changes) and in the revised draft of the registered report.

The virtual room acoustics is very unnatural. Room with identical absorption for walls ceiling and floor do not exist. This contradicts the high effort for generating the rooms using CATT. This is not a severe problem for monaural measurements. But I could imagine that the authors want to extend their focus on binaural measurements. For such measurements the RIR would be really problematic.

The rooms generated here are indeed very artificial. The goal of this experiment was not to test realistic rooms, but rather to be sure that the only varying parameter between the RIRs was the absorption coefficients of the room. In other words, with this configuration only the energy of the reflections of the speech and noise signals vary. The generalization of the results to more realistic conditions could be the subject of a follow-up study that could test the conditions of this experiment on realistic RIRs. The rooms were generated with CATT-Acoustic simply because it was available to us at this time.

It might be an issue that for each test condition 10 sentences containing the same pool of words are used. This depends on the manner of randomization mentioned below. This would have the consequence, that the later sentences in each block are biased by the words perceived in the previous trials as the best strategy is to repeat only words the listener hasn't heard so far in this list. In this case it is not a sentence intelligibility test but as a word spotting test combined with a short-term memory span test. This has not sufficient ecological validity to investigate the effects of reverberation and dip listening on sentence recognition.

In this study, the blocks and the sentences of each block are randomized. For example, the listener will receive the 2nd sentence of the 4th block, then the 8th sentence of the 2nd block, then the 3rd sentence of the 1st block... Thus, the memorization problem that you mention should not be an issue here.

The selection of the five fixed SNRs for measurements has not been specified clearly enough. In order to guarantee that NH and HI can be tested at the same SNRs, the intelligibility function of the test has to be known. As it is new material the intelligibility function is unknown, so far. As the single words have not been adjusted for equal intelligibility but for equal RMS, the slope of the intelligibility function can be expected to be flatter than in the FrMatrix test, see Kollmeier 2015 where the relation between slope and intelligibility of single words is explained. In order to specify the SNRs for testing the intelligibility function or at least the expected range has to be specified. I see that pilot measurements will be done, but it should be made clear which target intelligibilities should be met and if it is possible to fulfill the requirements of overlapping intelligibility ranges between conditions and listener groups.

To estimate the SNR range and the slope of the intelligibility function, we tested two NH listeners and one moderately impaired listener for the hardest and easiest conditions. We found a slope of 5%-10%/dB and a range between 0 dB and -20 dB for all the conditions for the NH listeners. For the HI listener we found that a range between 5 dB and -15 dB was necessary. These results were obtained with the revised version of the experiment (original sentences of the FrMatrix test, vocoded speech...).

These ranges are the ones that will be used in the experiment and are mentioned in the revised registered report.

P8 L7-8: what does it mean that the test condition is tested on a trial-by-trial order? Is a trial here a 10 sentence-list or a single sentence? I hope that the latter is meant. Then the memory span issue mentioned above is not an issue. But please specify how the tests are divided into blocks.

This point was not explained clearly, thanks for bringing this to our attention. It is indeed the latter: a trial is a single sentence. We tried to make this clearer in the new draft.

I also want to mention that there is evidence supporting the hypothesis that matrix sentences are less affected by reverberation than daily life speech (see for example Rennies et al 2014, 10.1121/1.4897398). More data are required concerning this question, but I think that this point should be discussed in this study.

Thank you again for bringing this to our attention. This point is now mentioned in the new draft of the registered report. Unfortunately, we do not see any way to avoid or compensate this robustness to reverberation, and we do not have any other French speech test that could be used here. We will try to address it in more details in the Discussion part of the article.

Minor comments:

P1 L52: white space in "the"

This has been corrected.

References are in French style, e.g. "Helfer et Wilber (1990)".

This has been corrected.

P2 L39-58: due to the limitation of the scale and the non-linear intelligibility function it is problematic to compare intelligibility scores in %. If an intelligibility score decreases, for instance, from 100 to 80% it is a much stronger effect compared to a decrease from 60 to 40%. Later it is mentioned that RAUs will be used to compensate for this. It might be required to take this into account already here. This also plays a role in P3 L12-13.

Indeed, the scores will be put in RAUs in the experiment presented here, but the results from other studies in the introduction were presented in percentage. We corrected this and put them in RAUs. This reduces the difference of effect of reverberation for NH and HI listeners in some previously published studies.

This also led us to notice a mistake in our power analysis: the simulations were done with the scores in percentage, instead of RAUs. When the RAU conversion is done, the difference between the two groups is so small that the Bayesian ANOVA confirms an absence of interaction RT x group 80 % of the time.

These elements led us to rethink our hypothesis 3-1. You can find more details about this above (in the summary of the main changes), and in the revised draft of the registered report.

P4 L40: Why has the dip listening benefit to be evaluated at comparable SNRs? This is only one aspect. You can also argue that the dip listening benefit has to be sufficient high sensation level in both groups in order to assure that audibility is not the limiting factor in the dips, especially for the HI listeners.

It is true that many studies investigating dip listening equalize audibility and SNR, such as Jensen and Bernstein (2019). However, equalizing audibility implies that only the supra-threshold elements are studied. The goal of this study was to study all the elements of hearing impairment, including the audibility loss. Thus, we decided to only correct the confounding effect of the SNR. It is important to note that to do this we use the NAL-RP amplification, which partly corrects audibility. Since the initial submission we also decided to add an audibility pre-test to make sure that the sentences were audible for all listeners.

P5 L53: Why do you plan to select the ear with the flatter hearing loss? One can also argue for selecting the better ear in order to reduce the effect of sensory deprivation due to the hearing loss of the weak ear.

The flat hearing loss criteria is indeed questionable. The main objective here is to stay close to the HI listeners tested by Harris and Swenson (1990), which had PTAs of 30-60 dB. If HI listeners have a more severe hearing loss, there is a risk that the NAL-RP amplification makes the signals uncomfortable. If they have a less severe hearing loss, there is a risk that there are not a lot of differences between the NH and HI listeners. The ear selected will be the one that is the closest to the PTAs of Harris and Swenson (1990). In the new draft of the registered report the flat hearing loss criteria has been removed.

In Beutelmann, Brand and Kollmeier (2010) different reverberation times and noises with different degrees of modulation have been tested with NH and HI listeners. The analysis is not as systematic as the analysis planned here, but there is at least one other study that investigated dip listening in reverberation in HI listeners.

We indeed omitted this study, thanks for bringing this to our attention, it is now mentioned in the introduction. However, since this study did not treat the confounding effect of SNR or equalized audibility, the dip listening benefit of the HI listeners is very low in all conditions and it is not really possible to draw any clear conclusions regarding the dip listening benefit decrease caused by reverberation.

P11: I think there is a typo in lower right field, second condition: "If $BF < 1/3$: the magnitude ..."

This has been corrected.

Appendix C

Dear editor,

We are resubmitting to you today the manuscript RSOS-210342.R2 intitled “Effects of reverberation on speech intelligibility in noise for hearing-impaired listeners” for consideration to be published in Royal Society Open Science. This is the stage 2 version of the approved protocol RSOS-210342.R1.

This study aims to better understand how reverberation affects intelligibility for hearing-impaired listeners. This should allow to understand more precisely which acoustic conditions are detrimental to them and, in the long term, it could help to improve the acoustic regulations related to room design. This work relies on a sufficient number of studies to propose precise hypotheses to be tested. It was thus appropriate to submit it as a registered report, in comparison to more prospective studies. Furthermore, submitting this manuscript as a registered report has add greater scientific value to the results, thanks to the reviewers’ inputs to the design of the study.

The approved Stage 1 protocol was archived on Zenodo under embargo. The data obtained and the codes used were archived under restricted access. The URL links appear in the manuscript at page 21. We confirm that no data for any pre-registered study was collected prior to the date of in principle acceptance.

Several supplementary analyses were deemed necessary to further investigate the results. They were also archived on Zenodo under restricted access. The URL link appear in the manuscript at page 21.

Best regards,
Raphael Cueille

Appendix D

Dear editor,

Thank you for your review and comments. We definitely agree with all of them.

First, we agree that the use of a sigmoid fitting was not mentioned at stage one of the registered report. At that stage we simply mentioned that we would apply a fitting on the results (ie the intelligibility scores in percent) to estimate the psychometric curves. Such a fitting is usually done using a sigmoid function as we did. This is now clearly stated in the manuscript.

The second important point is the RAU correction that is applied to the data. As planned at stage one, the RAU correction is applied after the fitting to get both the figures and the statistical analyses. This is also now clearly stated in the manuscript.

To summarize:

- 1- The intelligibility scores of the listeners are measured (in percentage).
- 2- Individual psychometric curves are fitted on these intelligibility scores (in percentage).
- 3- The individual psychometric fittings (in percentage) are converted in RAU and the statistical test are applied on the RAU scores.
- 4- The individual psychometric fittings (in percentage) are averaged across listeners in each group (averaging the curve parameters) and the mean curves are plotted on a RAU scale in order to visualize the group effects without being biased by floor or ceiling effects.

In order to clarify this point we have specified in the Procedure subsection that the fitting was done on the percentage scores using a sigmoid function, and that the RAU transformation was done after the fitting. We have also added footnotes in the tracked-change and clean versions to clarify this point.

At stage 2, the curves appearing in Figures 4, 5 and 6 were presented as the “mean psychometric curves”, which is actually very misleading. We have modified the captions in order to clearly indicate that we plot the mean of the RAU conversion of the psychometric curves.

Another concern of the reviewer was that the stage 2 version of the article indicated that the lower asymptote was allowed to vary between 0 and 0.05. This was a mistake: in fact, only the upper asymptote was allowed to vary. This has been corrected in the clean and tracked-change version.

Finally, the grammar errors highlighted by the reviewer have all been corrected.

In the tracked change version, the blue represents the modifications done between the stage 1 and stage 2 versions. The green represents the modifications done to take your last comments into account.